# FlexAC⚙️: Towards Flexible Control of Associative Reasoning in Multimodal Large Language Models

**Shengming Yuan**[1]* 
shengming.yuan@outlook.com

**Xinyu Lyu**[2]* 
xinyulyu68@gmail.com

**Shuailong Wang**[1] 
wslliongliong@gmail.com

**Beitao Chen**[1] 
chenbeitao@gmail.com

**Jingkuan Song**[3] 
jingkuan.song@gmail.com

**Lianli Gao**[1]† 
lianli.gao@uestc.edu.cn

[1]University of Electronic Science and Technology of China 
[2]Southwestern University of Finance and Economics, Chengdu, China 
[3]Tongji University

## Abstract

Multimodal large language models (MLLMs) face an inherent trade-off between *faithfulness* and *creativity*, as different tasks require varying degrees of *associative reasoning*. However, existing methods lack the flexibility to modulate this reasoning strength, limiting MLLMs' adaptability across factual and creative scenarios. To bridge this gap, we propose equipping MLLMs with mechanisms that enable *flexible control over associative reasoning*. We begin by investigating the internal mechanisms underlying associative behavior in MLLMs and **find that: (1)** middle layers play a pivotal role in shaping model's associative tendencies, **(2)** modifying representations in these layers effectively regulates associative reasoning strength, and **(3)** hallucinations can be exploited to derive steering vectors that guide this modulation. Building on these findings, we introduce **Flex**ible **A**ssociation **C**ontrol (**FlexAC**), a **lightweight and training-free** framework for modulating associative behavior in MLLMs. FlexAC first induces hallucination-guided intermediate representations to **encode associative directions**. Then, it selects high-association instances to construct effective associative steering vectors, whose strengths are **adaptively calibrated** to balance creative guidance with output stability. Finally, recognizing the multi-dimensional nature of associative reasoning, FlexAC incorporates task-specific associative vectors derived from a forward pass on a few target-domain samples, enabling models to follow **diverse** associative directions and better adapt to **creative tasks**. Notably, our method achieves up to a 5.8× improvement in creativity on Creation-MMBench and a 29% reduction in hallucination rate on CHAIR, surpassing existing baselines and demonstrating its effectiveness in enabling flexible control over associative reasoning in MLLMs. Our code is available at https://github.com/ylhz/FlexAC.

## 1 Introduction

In cognitive science, divergent and convergent thinking represent two distinct modes of human associative behavior: convergent thinking relies on typical, fact-based associations to support faithful

---

*Co-first author.

†Corresponding author.

39th Conference on Neural Information Processing Systems (NeurIPS 2025).

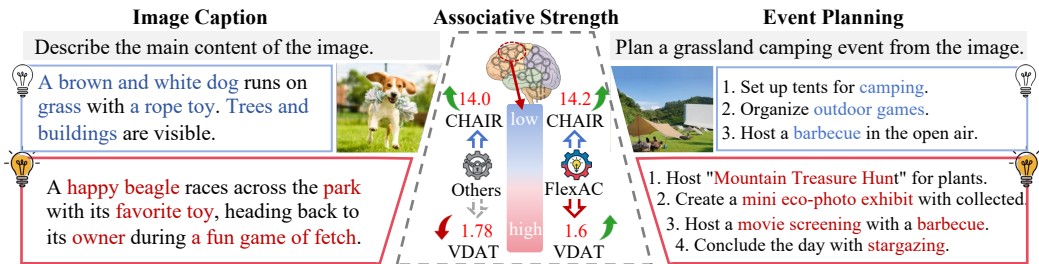

**Image Caption**
Describe the main content of the image.

A brown and white dog runs on grass with a rope toy. Trees and buildings are visible.

A happy beagle races across the park with its favorite toy, heading back to its owner during a fun game of fetch.

**Associative Strength**

14.0 CHAIR low 14.2 CHAIR

Others FlexAC

1.78 VDAT high 1.6 VDAT

**Event Planning**
Plan a grassland camping event from the image.

1. Set up tents for camping.
2. Organize outdoor games.
3. Host a barbecue in the open air.

1. Host "Mountain Treasure Hunt" for plants.
2. Create a mini eco-photo exhibit with collected
3. Host a movie screening with a barbecue.
4. Conclude the day with stargazing.

Figure 1: **Different tasks require different levels of associative reasoning**: factual tasks (*e.g.*, image caption) benefit from lower association, while creative tasks (*e.g.*, event planning) thrive on higher association. Existing methods suppress hallucinations at the cost of creativity (*e.g.*, -1.78 on VDAT; "Others" from Ha-DPO). FlexAC enables MLLMs to adjust associative reasoning strength accordingly.

reasoning, whereas divergent thinking engages atypical, context-dependent associations to foster creativity [1]. Recent studies show that multimodal large language models (MLLMs) [2, 3, 4] exhibit brain-like properties, such as structured embedding spaces [5], cross-modal integration [6], and higher-order cognitive functions [7], indicating that they emulate human associative processes. Consequently, like the human brain, **MLLMs require the capacity to flexibly regulate associative reasoning strength to support both faithful reasoning and creative generation.**

However, existing methods lack the flexibility to modulate associative reasoning strength, limiting MLLMs' adaptability across factual and creative scenarios. On one hand, current hallucination mitigation techniques, such as Contrastive Decoding [8, 9, 10] and Direct Preference Optimization [11], focus on improving faithfulness but often suppress associative reasoning capabilities, thereby hindering performance on tasks involving imaginative understanding and literary expression. On the other hand, how to enhance MLLMs' creativity in a controllable and task-specific manner remains underexplored. For instance, as illustrated in Figure 1, existing hallucination mitigation techniques improve faithfulness (14.0 ↓ in CHAIR) but lack mechanisms for enhancing creativity, resulting in reduced associative reasoning strength (1.78 ↓ in VDAT) and poor performance on tasks such as event planning. This gap highlights **the need for equipping MLLMs with controllable mechanisms to flexibly modulate associative reasoning strength based on task demands.**

To enable controllable modulation of associative reasoning strength, we begin by examining how associative behavior emerges within MLLMs. Drawing inspiration from prior works [12, 13], we hypothesize that hallucination and creativity arise from shared associative mechanisms, whose manifestations vary with task demands. To validate this, we collect input-response pairs containing both grounded (low-association) and hallucinated (high-association) outputs, and analyze their internal representations to uncover how associative behavior is reflected within the model. Our analysis reveals three key findings (see Section 2.1 and Section 2.2): (1) Associative behaviors are primarily encoded in the middle layers, where the representations of grounded and hallucinated responses become distinctly separable; (2) Modifying internal representations at these layers can effectively alter the strength and direction of associative reasoning; (3) Direction of hallucinated representations can stimulate associative reasoning capability, offering a potential control signal for this modulation. **These findings indicate that associative tendencies are encoded in middle layers and can be modulated through targeted interventions guided by hallucination.**

Motivated by these findings, we propose **Flex**ible **A**ssociation **C**ontrol (**FlexAC**), a lightweight and training-free framework for modulating associative behavior in MLLMs. The core idea is to first extract the associative vector from hallucinated responses (Phase I: Offline Control Vector Construction), which exhibit strong associative tendencies, and then apply it at inference time to guide model behavior (Phase II: Inference-Time Control). **In the Offline Control Vector Construction Phase**, FlexAC performs **three key steps:** (1) Hallucination-Guided Intermediate States: We collect grounded–hallucinated response pairs, and measuring the differences between their hidden states within model's middle layers, which encode the associative direction. (2) Instance Selection: To reduce noise from individual samples, we select the top-K response pairs with the largest association shifts and average their differences to obtain a reliable steering vector. (3) Directional Integration: To further support tasks requiring multi-dimensional associations (e.g., storytelling or metaphor generation), we augment the general associative vector with task-specific associative vectors derived

from GPT-4o-generated, high-association samples. These vectors are incorporated at inference time for fine-grained and controllable modulation. **In the Inference-Time Control Phase**, we apply the combined steering vector during inference. However, uniformly applying this vector can lead to over-steering, especially for inputs already exhibit strong associative behavior, causing irrelevant outputs or stylistic drift. To mitigate this, we introduce **Steering Intensity Calibration**, which adaptively scales the steering vector: amplifying it when associative behavior is weak, and attenuating it when the desired level has been reached.

To evaluate the effectiveness of FlexAC in controlling associative behavior, we conduct experiments across three fronts: hallucination mitigation (CHAIR [14] and POPE [15] for low-association tasks), creativity enhancement (VDAT and Creation-MMBench [16] for high-association tasks), and general-purpose evaluation (MME [17], MMMU [18], and MMStar [19]). Results show that FlexAC enables flexible modulation of associative reasoning capability, achieving state-of-the-art performance on both low- and high-association tasks while enhancing general capabilities.

In summary, our contributions are fourfold: (1)We present a unified perspective that links hallucination and creativity to associative reasoning, identifying middle-layer representations as key control points. (2)We propose **FlexAC**, a lightweight and training-free framework for flexible modulation of associative strength, enabling task-aware switching between hallucination suppression and creativity enhancement. (3)We introduce VDAT, a benchmark specifically designed to evaluate associative reasoning strength. (4)We conduct comprehensive experiments demonstrating that FlexAC effectively controls associative behavior across hallucination, creativity, and general-purpose benchmarks.

## 2 Analyzing and modulating associative behavior in MLLMs

### 2.1 Analyzing layer-wise localization of associative processes

**Feature Distance Analysis: Quantifying layer-wise differences between associative and non-associative representations.** To identify where associative behavior emerges, we analyze layer-wise representations in LLaVA-1.5-7b using 1000 images from COCO2024. For each image, we collect two type of responses: a grounded (non-associative) response from the model's default output, and a hallucinated (associative) response induced via blurred inputs and specific prompts [8]. Here, we use hallucinated responses to represent associative behavior, as they often include many imaginative contents, objects that do not exist in the image but are semantically related to the scene, reflecting the model's associative tendencies. We then extract the associative features $f^{(a)}$ and non-associative features $f^{(n)}$ from all intermediate layers for both data types (visualized in Figure 4). The full data construction and feature extraction process is detailed in Appendix B. Next, we compute the cosine distance and Euclidean distance between $f^{(a)}$ and $f^{(n)}$ across all layers. The cosine distance $\mathcal{D}_{\cos}$ is used to evaluate the directional alignment between associative and non-associative features, while Euclidean distance $\mathcal{D}_{\text{Euc}}$ measures the spatial distribution differences.

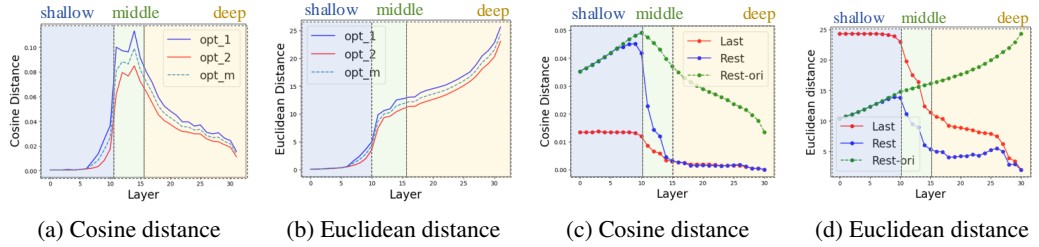

(a) Cosine distance    (b) Euclidean distance    (c) Cosine distance    (d) Euclidean distance

Figure 2: (a) and (b) show the cosine and Euclidean distances between associative and non-associative features across layers. (c) and (d) illustrate the impact of replacing associative features in different layers on subsequent layers."Last" and "Rest" denotes the final layer difference $d_L$ and the average layer difference $\bar{d}_{m:L}$, respectively. "Rest-ori" represents the original mean feature distance $\bar{d}_{m:L}$ without replacement.

As shown in Figures 2a and 2b, both cosine and Euclidean distances remain consistently low in the **shallow layers** (layers 0–9), indicating shared low-level perception. However, for middle and deep layers, we observe distinct patterns between cosine and Euclidean distance when comparing grounded

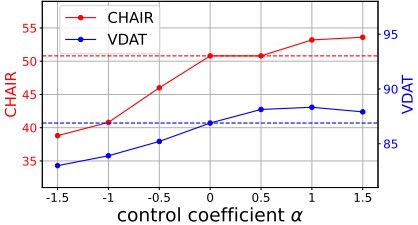 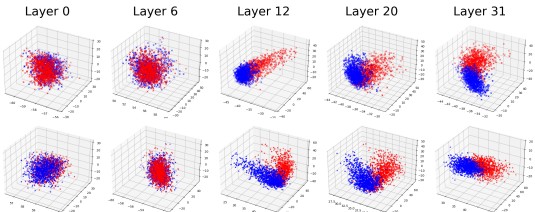

Figure 3: **Impact of Middle Layer Control on Hallucination-Driven Behavior.** Adjusting $\alpha$ increases both hallucination (CHAIR) and creativity (VDAT), suggesting that associative strength can be modulated through middle-layer control using hallucination representations.

Figure 4: **Visualization of feature representations in LLaVA-1.5-7b**, reduced via PCA, shows red (associative) and blue (non-associative) points. The feature distributions show increasing separation in deeper layers, illustrating how associative distinctions are formed. See Appendix F.1 for all layers.

and hallucinated responses across layers. Cosine distance peaks in the **middle layers** (layers 10–15), indicating that this stage is where feature directions diverge most significantly—suggesting that associative behavior is primarily introduced and shaped in this range. In contrast, Euclidean distance increases steadily across both **middle and deep layers** (layers 10–31), implying that the overall feature magnitudes continue to drift even in later stages. This discrepancy raises **a key question:** *Is associative behavior actively introduced in the **deep layers**, or are these differences merely the propagated result of associative shifts originating in the **middle layers**?*

**Layer Intervention: Verifying the source of associative signals.** To answer this, we conduct a layer intervention experiment (Figure 5), in which we replace the associative feature $f_m^{(a)}$ with the corresponding non-associative feature $f_m^{(n)}$ at different layers $m$, and observe the influence on downstream representations. The modified feature propagation is defined as:

$$f_l^{\text{modified}} = \begin{cases} f_l^{(a)} & l < m \\ \mathcal{M}^l \circ \cdots \circ \mathcal{M}^{m+1}(f_m^{(n)}) & l \geq m, \end{cases}$$
(1)

where $\mathcal{M}^l$ denotes the $l$-th layer of the model. We evaluate the impact by calculating the final layer difference $d_L$ and the average layer difference $\bar{d}_{m+1:L}$ as follows:

$$d_L = \mathcal{D}(f_L^{\text{modified}}, f_L^{(n)})$$
(2)

$$\bar{d}_{m:L} = \frac{1}{L-m} \sum_{i=m+1}^{L} \mathcal{D}(f_i^{\text{modified}}, f_i^{(n)}),$$
(3)

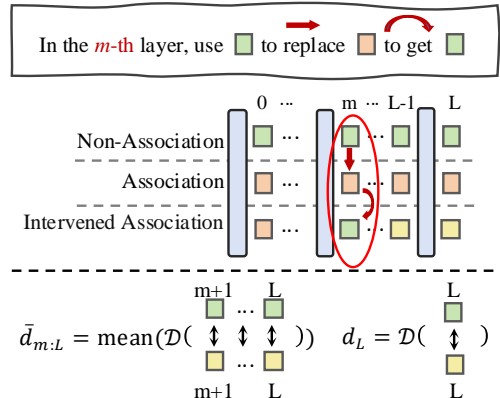

$$\bar{d}_{m:L} = \text{mean}(\mathcal{D}(\ )) \quad d_L = \mathcal{D}(\ )$$

Figure 5: **Layer Intervention for Association Localization.** The goal is that locating the key layers for associative feature generation. Associative features are replaced with non-associative ones at different layers, and the impact on subsequent layers is evaluated using $d_L$ and $\bar{d}_{m:L}$.

where $\mathcal{D}(\cdot)$ denotes either cosine or Euclidean distance.

Results in Figures 2c and 2d show that replacing features in shallow layers (layers 0-9) leads to minimal changes in downstream representations, indicating limited influence on associative processing. In contrast, replacing features in middle layers (layers 10-15) significantly reduces divergence in later layers, suggesting that these layers are the primary source of associative behavior. Replacements in deep layers (layers 16-31) again have limited impact, implying that these layers mainly propagate rather than generate associative features. More visualization in Appendix F.2.

**Finding 1**: Middle layers are critical for shaping MLLM's associative behavior.

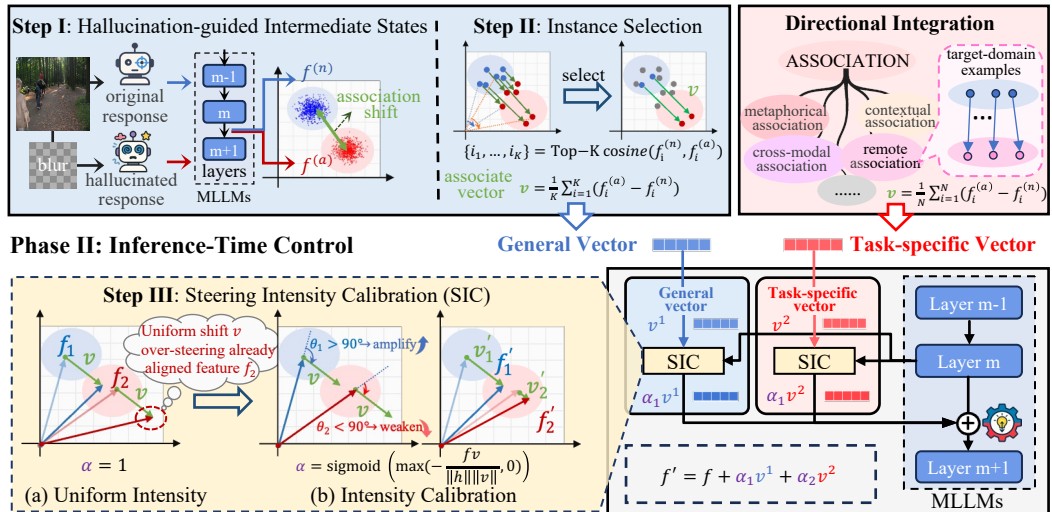

Figure 6: **Overview of the proposed FlexAC framework. Phase I: Offline Control Vector Construction** extracts a **general associative vector** from hallucination-guided intermediate features (Step I), by selecting Top-K instance pairs with maximal association shifts (Step II). It also constructs **task-specific associative vectors** from a few target-domain examples (Step III), reflecting diverse associative needs. **Phase II: Inference-Time Control** injects these vectors into middle-layer features. A Steering Intensity Calibration (SIC) module adaptively adjusts the influence of each vector per sample to achieve controllable associative reasoning strength.

## 2.2 Analyzing control strategies for associative behavior modulation

This analysis investigates whether associative behavior can be modulated by manipulating middle-layer representations, and whether hallucinated responses reveal effective directions for such control. Using the same grounded and hallucinated feature pairs from Section 2.1, we compute feature differences layer by layer to derive the control direction:

$$v_l = f_l^{(a)} - F_l^{(n)}. \tag{4}$$

We then apply this steering vector during inference to modulate the model's output by adjusting the middle-layer features with control coefficient $\alpha$:

$$f_l^{\text{control}} = f_l + \alpha \cdot v_l. \tag{5}$$

To assess the impact of steering on associative behavior, we introduce VDAT (Visual-Divergent Association Test), a benchmark that evaluates a model's associative reasoning by prompting it to generate unrelated nouns to the input image, thereby measuring its capacity for visual-driven divergent thinking (details in Section 3.1). As shown in Figure 3, increasing $\alpha$ from -1.5 to 1.5 raises CHAIR from approximately 38.8 to 53.6 and VDAT from around 83 to 87.9, indicating that higher $\alpha$ values lead to both more hallucination and stronger associative ability. Conversely, decreasing $\alpha$ reduces both scores. These results highlight that $\alpha$ provides a controllable mechanism for modulating associative behavior in MLLMs. These results yield two key findings:

> **Finding 2**: Modifying middle layers enables control over associative reasoning strength.
> **Finding 3**: Hallucinations help derive steering vectors to guide associative reasoning.

## 2.3 Flexible association control

Based on our findings in Sections 2.1 and 2.2, we propose **Flexible Association Control (FlexAC)**, a lightweight, training-free framework for modulating associative behavior in MLLMs. As illustrated in Figure 6, FlexAC operates in two phases: (I) **Offline Control Vector Construction**, which derives general and task-specific associative directions, and (II) **Inference-Time Control**, which injects these directions into middle-layer features for dynamic modulation.

**Phase I: Offline Control Vector Construction.** To capture a general associative direction, we first induce hallucinated responses that exhibit high associative behavior (**Finding 3**). For each input, we extract hidden features from the middle layer $l$, where associative distinctions are most prominent (**Finding 1**), resulting in paired features $f_l^{(a)}$ and $f_l^{(n)}$. We select the top-$K$ pairs with the highest cosine distances to construct a representative direction vector:

$$\mathcal{I} = \text{Top} - \text{K}\left(\mathcal{D}_{\cos}(f_{l,i}^{(a)}, f_{l,i}^{(n)})\right); v_l = \frac{1}{|\mathcal{I}|}\sum_{i \in \mathcal{I}}\left(f_{l,i}^{(a)} - f_{l,i}^{(n)}\right) \qquad (6)$$

To handle tasks requiring diverse associative patterns (*e.g.*, metaphorical, contextual), we further construct task-specific associative vectors from a few high-association, instruction-aligned examples. As vanilla MLLMs struggle to produce such outputs, we leverage GPT-4o to generate high-quality associative outputs.

**Phase II: Inference-Time Control.** During inference-time phase, we adjust the hidden state $f_l$ at middle layer $l$ (**Finding 2**) by injecting a combination of general associative vector $v_l^{\text{gen}}$ and task-specific associative vector $v_l^{\text{task}}$:

$$f_l^{\text{control}} = f_l + \alpha_{\text{gen}} \cdot v_l^{\text{gen}} + \alpha_{\text{task}} \cdot v_l^{\text{task}} \qquad (7)$$

where $\alpha$ is the tunable coefficient that controls the steering intensity. This formulation is grounded in recent theoretical findings [20], which reveal that task-specific differences in model weights exhibit linearly decomposable structures. This property supports our assumption that associative directions can be independently extracted and combined within the hidden space.

However, directly applying a uniform steering vector across all inputs can lead to over-steering, especially when the input already exhibits strong associative behavior, causing deviation from the intended semantic space (see Step III of Figure 6). To mitigate it, we introduce steering intensity calibration strategy, which adjusts the steering strength $\alpha$ based on:

$$\alpha = \text{sigmoid}\left(\max\left(-\frac{f_l \cdot v_l}{\|f_l\|\|v_l\|}, 0\right)\right) \qquad (8)$$

This formulation increase steering strength when the current representations is misaligned with the associate direction, and suppresses it when already aligned. We further normalize the modulated feature to preserve its scale:

$$f_l^{\text{control}} \leftarrow f_l^{\text{control}} \cdot \frac{\|f_l\|}{\|f_l^{\text{control}}\|} \qquad (9)$$

This mechanism enables precise, interpretable modulation of associative behavior, allowing MLLMs to shift smoothly between factual accuracy and creative generation (Figure 8).

## 3 Experiments

### 3.1 Experimental setup

**Evaluation Metric**: To evaluate the effectiveness of FlexAC, we conduct experiments on three benchmark types: (1) **hallucination**, using CHAIR [14] and POPE [15] to assess object-level factual consistency; (2) **creativity**, using our proposed VDAT for associative reasoning and Creation-MMBench [21] for open-ended image-grounded generation; and (3) **general-purpose capability**, using MME [17], MMMU [18] and MMStar [19] to ensure core perception and reasoning are preserved. Metric details are in Appendix C.

*VDAT: Visual Divergent Association Test.* To measure a model's associative reasoning and creative potential more directly, we introduce VDAT, a diagnostic benchmark that complements Creation-MMBench by focusing specifically on associative reasoning strength. Inspired by [22], VDAT prompts the model to generate multiple nouns that are unrelated both to the input image, capturing its capacity for visual-driven divergent thinking (Figure 7). The metric is computed using CLIP ViT-L/14 embeddings.

**Implementation Details.** We evaluate the effectiveness of our FlexAC on LLaVA-1.5 [23], Qwen-VL [3], and Deepseek-VL [4], comparing it with Ha-DPO [11], VCD [8], and VAF [24]. From the COCO2014 [25] dataset, we randomly selected 2000 images and then applied Instance Selection

to choose 50 images for generating the general association vector. For the layer intervention, we manipulated the following layers based on each model's associative strength: Qwen-VL (layers 15, 16, 17), LLaVA-1.5 (layers 11, 12, 13), and Deepseek-VL (layers 4, 5, 6). For FlexAC-P (faithfulness-enhanced) and FlexAC-C (creativity-enhanced), the control coefficient $\alpha$ is set to -1 and 1, respectively. All experiments were conducted on 8×RTX 4090 GPUs. The parameter analysis of the number of images is provided in Appendix E.1.

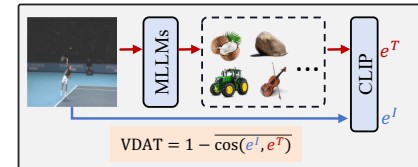

Figure 7: **Visual Divergent Association Test (VDAT)** evaluates a model's associative reasoning by prompting it to generate unrelated nouns from an image, and quantifies performance through image-text measured using CLIP embeddings.

Table 1: **Performance on hallucination benchmarks.** FlexAC here denotes the version configured to suppress associative behavior, aiming to improve factual accuracy (faithfulness).

| Models | Methods | CHAIR | | | | POPE | | | |
|---|---|---|---|---|---|---|---|---|---|
| | | CHAIR$_S$ ↓ | CHAIR$_I$ ↓ | Recall | Len | F1-score ↑ | Accuracy ↑ | Precision ↑ | Recall |
| Qwen-VL | Regular | 40.6 | 12.5 | 71.7 | 94.6 | 85.6 | 86.6 | **92.9** | 79.3 |
| | VCD | 42.0 | 11.2 | 71.7 | 91.2 | 86.3 | 87.2 | 92.4 | 81.0 |
| | VAF | 38.0 | 11.7 | 72.2 | 91.4 | 86.5 | 87.2 | 91.4 | 82.0 |
| | FlexAC (Ours) | **19.2** | **5.4** | 62.5 | 74.8 | **87.1** | **87.4** | 89.3 | 85.1 |
| LLaVA-1.5 | Regular | 50.8 | 14.3 | 79.7 | 97.3 | 86.5 | 87.2 | 91.5 | 82.0 |
| | Ha-DPO | 36.8 | **10.4** | 74.0 | 88.3 | 83.9 | 85.3 | **92.6** | 76.7 |
| | VCD | 51.0 | 15.5 | 79.1 | 98.9 | 84.3 | 84.9 | 88.1 | 80.7 |
| | VAF | 47.8 | 13.7 | 79.2 | 96.1 | 86.9 | 87.1 | 87.9 | 85.9 |
| | FlexAC (Ours) | **36.6** | **10.4** | 75.0 | 95.1 | **87.9** | **87.8** | 87.1 | 88.8 |
| Deepseek-VL2 | Regular | 32.6 | 9.2 | 67.0 | 121.0 | 88.5 | 88.4 | 88.1 | 88.8 |
| | VCD | 36.6 | 11.3 | 67.2 | 128.2 | 87.9 | 87.8 | 87.6 | 88.1 |
| | VAF | 32.0 | 9.2 | 66.2 | 119.0 | 88.5 | 88.4 | 87.6 | 89.4 |
| | FlexAC (Ours) | **28.6** | **8.1** | 64.7 | 117.0 | **88.6** | **88.5** | 88.4 | 88.7 |

## 3.2 Main results

**Results on Hallucination Benchmark.** To evaluate FlexAC's ability to improve factual accuracy in faithfulness-focused tasks, we conduct experiments on CHAIR and POPE. To this end, we set $\alpha$ in FlexAC to 1, selecting the precision-optimized variant. As shown in Table 1, FlexAC consistently achieves the lowest hallucination scores on most models and metrics. For examples, on CHAIR$_S$, FlexAC reduces hallucination to 19.2 (↑21.4) on Qwen-VL, 36.6 (↑14.2 vs. Regular) on LLaVA-1.5, and 28.6 (↑4.0) on Deepseek-VL2. On CHAIR$_I$, it similarly achieves the best scores (5.4, 10.4, and 8.1 respectively). In terms of POPE accuracy, FlexAC achieves the highest F1-score on LLaVA-1.5 (87.9) and comparable or superior precision and recall across the board. These results highlight FlexAC's ability to flexibly suppress excessive associative behavior in factual tasks, leading to improved accuracy across models.

**Results on Creativity Benchmark.** To evaluate FlexAC's ability to enhance associative reasoning in creative tasks, we conduct experiments on VDAT(Table 2) and Creation-MMBench(Table 3).

As shown in Table 2, hallucination mitigation methods like Ha-DPO reduce hallucinations but impair associative capacity, leading to lower creativity (*e.g.*, VDAT score of 85.11 vs.

Table 2: **Performance on VDAT.** FlexAC here denotes the version optimized to enhance associative behavior for creative tasks (creativity).

| Methods | Qwen-VL | LLaVA-1.5 | DeepSeek-VL2 |
|---|---|---|---|
| Regular | 84.85 | 86.89 | 84.54 |
| Ha-DPO | - | 85.11 | - |
| VCD | 83.69 | 86.83 | 84.62 |
| VAF | 84.95 | 86.79 | 84.61 |
| FlexAC (Ours) | **86.58** | **88.49** | **84.76** |

86.89 for the regular model). In contrast, FlexAC improves remote associative reasoning, achieving a higher VDAT score of 88.49. To further verify the validity of the VDAT metric, we conduct a user study presented in Appendix D.1. Further, on Creation-MMBench Table 3, we report VFS (Visual Fidelity Score), which evaluates image-text alignment, and Reward, which quantifies creativity

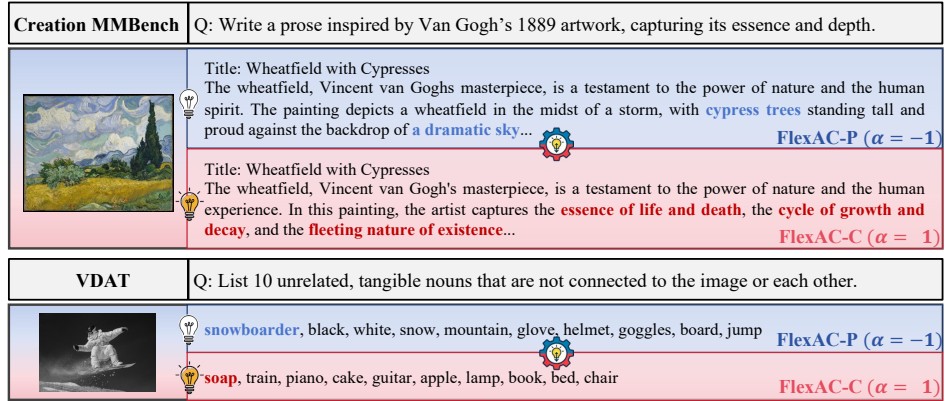

Figure 8: **Visualization of FlexAC's Control on Associative Reasoning.** This figure illustrates example outputs from Creation MMBench and VDAT, showing how FlexAC-P (faithfulness) and FlexAC-C (creativity) adjust the level of associative strength in the generated responses.

improvements relative to the base model (Qwen-VL). FlexAC achieves the highest Reward (10.92), outperforming methods like VCD (-3.86) and VAF (-1.63), while maintaining competitive VFS.

Qualitative examples in Figure 8 further support this: in Creation-MMBench, FlexAC-P focuses on concrete visual elements (e.g., "cypress trees"), while FlexAC-C introduces abstract themes (e.g., "life and death"). In VDAT, FlexAC-P outputs image-relevant nouns (e.g., "snowboarder"), whereas FlexAC-C generates semantically distant words (e.g., "guitar", "apple"), demonstrating enhanced divergent thinking. These examples confirm that FlexAC effectively modulates associative strength to meet diverse creative demands. For additional examples, see Appendix F.3.

Table 3: **Performance on Creation-MMBench.** We report results on four subcategories: Literary Writing (LW), Common Functional Writing (CFW), Professional Functional Writing (PFW), and Creative Multimodal Understanding (CMU). FlexAC here denotes the version optimized to enhance associative behavior for creative tasks (creativity).

| Methods | Overall | | LW | | CMU | | PFW | | CFW | |
|---|---|---|---|---|---|---|---|---|---|---|
| | VFS | Reward | VFS | Reward | VFS | Reward | VFS | Reward | VFS | Reward |
| Regular | 6.10 | 0.00 | 6.83 | 0.00 | 5.53 | 0.00 | 5.58 | 0.00 | 6.66 | 0.00 |
| VCD | 6.05 | -3.86 | 6.68 | -2.71 | 5.67 | 2.50 | **5.61** | -3.77 | 6.46 | -6.57 |
| VAF | 6.06 | -1.63 | 6.39 | -3.96 | 5.57 | -4.17 | **5.61** | -0.53 | 6.64 | -0.93 |
| FlexAC (Ours) | **6.25** | **10.92** | **7.20** | **15.63** | **5.83** | **6.11** | 5.43 | **5.96** | **7.00** | **15.65** |

**Results on General-Purpose Benchmark.** To evaluate the generalization capabilities of FlexAC across a range of tasks, we conduct experiments on three standard multimodal benchmarks using Qwen-VL: MME, MMMU, and MMStar. These benchmarks cover a wide range of capabilities including fine-grained grounding, reasoning, and instruction following.

As shown in Figure 9, both FlexAC-P (faithfulness-enhanced) and FlexAC-C (creativity-enhanced) maintain performance similar to the vanilla model across most categories, indicating no significant compromise in general capabilities. Notably, FlexAC-C outperforms the baseline on the OCR task in MME, likely due to its enhanced ability to associate text with related visual entities, improving inference and disambiguation under challenging conditions.

## 3.3 Ablation study

**Layer-wise Control Analysis.** We investigate the impact of middle layers on associative reasoning and identify the optimal control layers by testing interventions on shallow, middle, and deep layers, evaluating their effects on both CHAIR and VDAT metrics.

The results in Figure 10 demonstrate that middle layers have the most significant impact on performance: FlexAC-P achieves the best CHAIR results when suppressing associative behavior, while FlexAC-C shows the highest VDAT scores when enhancing creativity. In contrast, controlling shallow

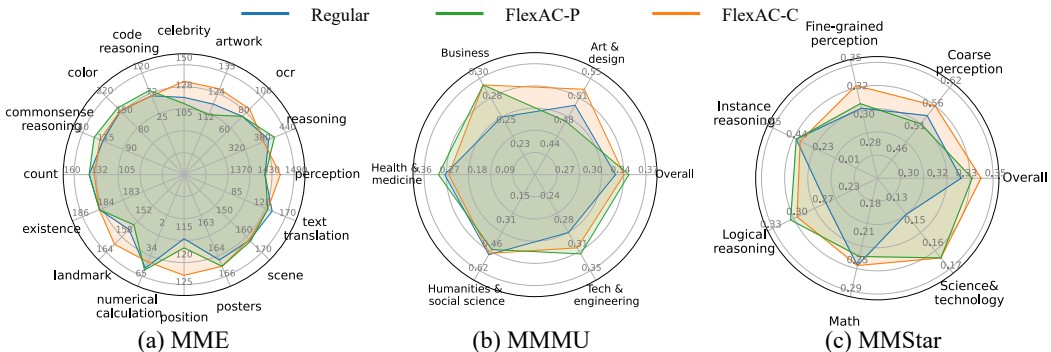

(a) MME       (b) MMMU       (c) MMStar

Figure 9: **Performance on general-purpose benchmarks.** Comparison of Regular, FlexAC-P (faithfulness-enhanced, $\alpha = -1$), and FlexAC-C (creativity-enhanced, $\alpha = 1$).

or deep layers has minimal effect. Based on these findings, we select layers 15, 16, and 17 as the control layers for Qwen-VL; results for other models are provided in Appendix E.2.

**Effectiveness of different Components.** We conducted an ablation study to assess the impact of components within FlexAC, including Instance Selection (IS), Steering Intensity Calibration (SIC), and Directional Integration (DI), on faithfulness (CHAIR) and creativity (VDAT).

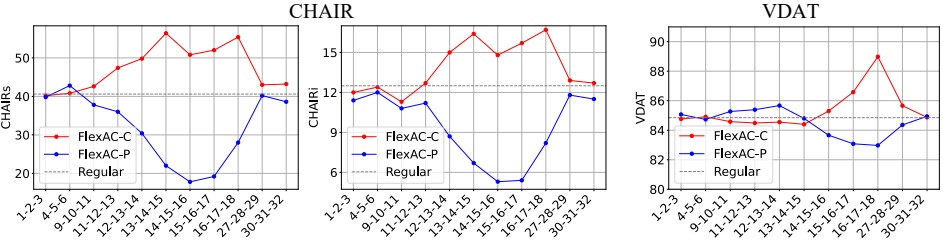

Figure 10: **Layer-wise analysis of control effectiveness in FlexAC.** The x-axis represents the control layers, while the y-axis shows the performance of the model on CHAIR and VDAT metrics.

As shown in Figure 11, for CHAIR, FlexAC(P) achieves the lowest $CHAIR_S$ score (19.2), indicating effective hallucination reduction compared to the regular model (40.6). When IS and SIC are removed from FlexAC (FlexAC-IS-SIC), performance slightly worsens (30.4), confirming their role in enhancing faithfulness. Similarly, for creativity, FlexAC-C scores the highest VDAT (86.58). Removing IS and SIC in FlexAC-IS-SIC leads to a small decrease (85.05), while FlexAC-DI results in a slight improvement, highlighting the importance of DI for creativity. In summary, FlexAC enables flexible adjustment of associative strength to meet the needs of different tasks, balancing hallucination reduction and creativity enhancement effectively.

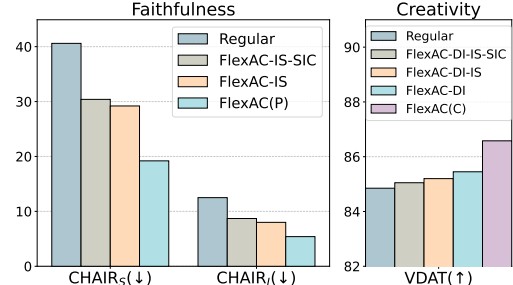

Figure 11: **Ablation study on components**, showing the impact of Instance Selection (IS), Steering Intensity Calibration (SIC), and Directional Integration (DI).

## 4 Related work

**Multimodal Large Language Models.** Recent advances in large language models (LLMs) [26, 27, 28] have led to the emergence of multimodal LLMs (MLLMs) that incorporate visual inputs for enhanced capabilities [29, 30]. LLaVA [2, 23] improves instruction-following via visual instruction tuning, while Qwen-VL [3] enhances spatial reasoning through visual grounding. DeepSeek-dVL2 [4] adopts a Mixture-of-Experts architecture to improve multimodal comprehension.

**Hallucination in MLLMs.** MLLMs still face various safety risks [31, 32, 33]. Hallucination is one of the core challenges, where MLLMs generate content misaligned with visual input [34]. To address

this, VCD [8] employs contrastive decoding, and VAF [24] enhances visual signal processing during fusion. HA-DPO [11] reduces hallucinations via preference optimization.

**Creativity in Large Models.** Creativity, involving divergent thinking and novel associations [35, 36], has been explored in LLMs via cognitive theories like dual-pathway [37]. Olson *et al*. [38] and Chen & Ding [39] promote remote associations; MacGyver [40] and CLOT [41] tackle functional fixedness and divergent thinking, respectively. Creation-MMBench [21] provides a benchmark for evaluating image-grounded creative generation.

## 5 Conclusion

In this work, we investigate the root of associative behavior in MLLMs, finding that middle-layer representations govern associative reasoning strength and that hallucinated responses encode reliable steering directions. Based on these insights, we propose FlexAC, a lightweight, training-free framework that combines hallucination-guided steering with adaptive calibration and in-context augmentation. FlexAC enables controllable creativity and achieves state-of-the-art performance across hallucination, creativity, and general-purpose benchmarks. **Limitations:** FlexAC requires white-box access to hidden states and is not applicable to black-box models like ChatGPT.

## 6 Acknowledgements

This study is supported by grants from the National Natural Science Foundation of China (Grant No. U23A20315, No. 62425208, No. U22A2097, No. 62122018, No. 62020106008), Shenzhen Science and Technology Program (No.JCYJ20240813114208012), Fundamental Research Funds for the Central Universities, and Natural Science Foundation of Sichuan Province (Grant No. 2025ZNSFSC1463).

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

# FlexAC⚙: Towards Flexible Control of Associative Reasoning in Multimodal Large Language Models (Supplementary Material)

## CONTENTS

## A  Broader Impacts

FlexAC introduces finer control over the associative behavior of MLLMs, enabling safer and more context-appropriate responses across tasks. This may benefit applications requiring factual precision (e.g., education, medical support) or creative output (e.g., storytelling, art generation). However, enhancing associative capacity also increases the model's expressive power, which—if misused—could lead to persuasive but unfounded generations. As with all generation-controlling techniques, FlexAC should be deployed alongside robust safeguards to ensure alignment with human intent and ethical use.

## B  Data Generation and Feature Extraction

**Inducing and Representing Model Associations**: To investigate the causes of model association, we generate two data distributions: one from the model's original outputs (non-associative) and another with induced associative content using blurred images and tailored prompts [8, 42]. For example, the model is prompted with: "Describe the image and include some hallucinated objects that are imagined but do not exist in the image, as if they were real." Following [12], we construct a multiple-choice dataset to capture feature distributions. The model is given an image and prompted to generate detailed responses, with two predefined options (Figure 12): [1] non-associative (factual) and [2] associative (creative). The hidden states corresponding to these inputs are extracted to obtain distinct feature representations, $F_{\text{non-assoc}}^l$ and $F_{\text{assoc}}^l$, capturing the model's internal response to both associative and non-associative prompts across different layers.

> ```
> <image>
> ```
> **Question:** Please describe this image in detail.
> [1] responses **without** association
> [2] responses **with** association
> Please select the most appropriate answer: [1 or 2]

Figure 12: The prompt for extracting associative and non-associative features

## C  Metrics details

All comparative experiments are conducted using the VLMEvalKit[3]. For binary choice questions, we prompt the model with: "Please answer Yes or No." We evaluate three models in our experiments: LLaVA-1.5 (`liuhaotian/llava-v1.5-7b`), Qwen-VL (`Qwen/Qwen-VL-Chat`), and DeepSeek-VL2 (`deepseek-ai/deepseek-vl2-tiny`).

**VDAT**  : VDAT fills a gap in evaluating the creative potential of multimodal models, which previous metrics did not adequately address. To ensure consistency, both CHAIR and VDAT were evaluated using the same 500 images, randomly selected from the MSCOCO dataset.

**Creation-MMBench**  [16]: Creation-MMBench is a multimodal benchmark designed to evaluate the creative capabilities of MLLMs in real-world, image-grounded scenarios. It contains 765 test cases across 51 fine-grained tasks, with instance-specific criteria that assess both imaginative quality and visual consistency. In contrast to prior work that compares models to GPT-4o, our evaluation focuses on measuring improvements over each model's own vanilla baseline.

**CHAIR**  [14]: Caption Hallucination Assessment with Image Relevance (CHAIR) is a metric designed to evaluate the hallucination of image caption task. It measures the hallucination rate of the generated text by comparing the generated caption with the ground-truth caption. CHAIR consists of

---

[3]https://github.com/open-compass/VLMEvalKit

two metrics: CHAIR$_S$ and CHAIR$_I$. They can be calculated as follows:

$$\text{CHAIR}_S = \frac{|\{hallucinated\ objects\}|}{|\{all\ mentioned\ objects\}|}, \tag{10}$$

$$\text{CHAIR}_I = \frac{|\{captions\ with\ hallucinatedobjects\}|}{|\{all\ captions\}|}. \tag{11}$$

**POPE** [15]: The Polling-based Object Probing Evaluation (POPE) is a metric developed to evaluate object hallucination in MLLMs. By framing the evaluation as a series of Yes-or-No questions about specific objects in images, POPE avoids issues related to instruction sensitivity. Using three sampling strategies—Random, Popular, and Adversarial—it effectively examines models' tendencies to hallucinate frequently occurring or co-occurring objects, providing a stable and reliable assessment of object hallucination. Refer to [15], we built POPE on 500 randomly selected MSCOCO [25] validation images, each containing over three ground-truth objects and six constructed questions.

**MME** [17]: MLLM Evaluation benchmark (MME) is a benchmark designed to assess multimodal large language models (MLLMs) across core skills in perception and cognition, such as object recognition, attribute identification, reasoning, and translation. Using accuracy-based metrics, MME provides objective insights into model capabilities, highlighting areas for improvement in understanding and reasoning.

**MMMU** [18]: MMMU (Massive Multi-discipline Multimodal Understanding and Reasoning) is a large-scale benchmark targeting expert-level multimodal understanding and reasoning. It comprises 11.5K college-level questions across 6 disciplines and 30 subjects, featuring 30 diverse image types such as charts, medical scans, diagrams, and chemical structures. MMMU emphasizes deep domain knowledge and deliberate reasoning, challenging models to integrate perception, knowledge, and logic in complex tasks. It serves as a necessary testbed for evaluating progress toward Expert AGI.

**MMStar** [19]: MMStar is a high-quality benchmark designed to evaluate vision-language models on truly vision-dependent tasks. It includes 1,500 human-curated samples across 6 core capabilities and 18 fine-grained skills, ensuring minimal data leakage and strong visual grounding.

# D  Detailed Experimental Results

## D.1  User study

To validate the effectiveness of the VDAT metric as a measure of associative creativity, we conducted a human evaluation study comparing FlexAC against several baselines. Specifically, we randomly selected 30 image-response examples from the Qwen-VL evaluation set and presented them to 15 human raters. For each example, two responses were shown—one from FlexAC and one from a baseline method (Regular, VAF, or VCD). Participants were asked to judge which response contained objects more unrelated to the image, as a proxy for stronger remote association. The response options were presented as "Answer A" and "Answer B," with the method-to-label mapping randomized in each trial to eliminate bias. Raters evaluated each pair on a five-point scale ranging from "A is much better than B" to "B is much better than A." These choices were then converted to numeric scores for aggregation—for example, "$A >> B$" assigns 3 points to A, "$A = B$" assigns 1 point to both A and B.

Figure 13 shows that FlexAC consistently receives higher average scores than all baselines, with low variance across users. Figure 14 further reveals that over 70% of responses favored FlexAC ($A > B$ or $A >> B$), while fewer than 6% favored the baseline. These results demonstrate strong alignment between the VDAT metric and human judgment. Figure 15 provides a screenshot of the evaluation interface. Together, these findings support VDAT as a valid and human-aligned metric for measuring associative creativity in vision-language generation tasks.

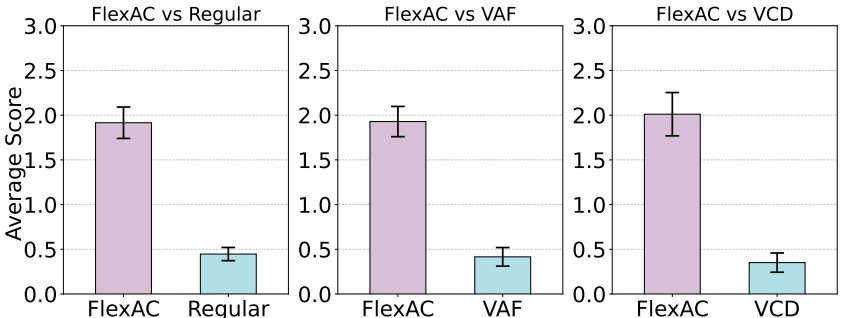

Figure 13: Average user ratings comparing FlexAC with baseline methods on the VDAT task. Each bar represents the average score across 15 users for 30 randomly selected image-response pairs. Error bars indicate the maximum and minimum individual user scores, reflecting rating consistency. Higher scores indicate stronger perceived remote association ability.

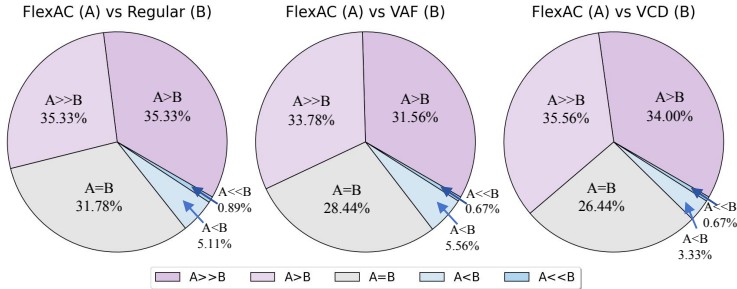

Figure 14: Distribution of user rating preferences when comparing FlexAC with each baseline on the VDAT task. $A = B$ indicates equal preference; $A >> B$ and $A > B$ mean FlexAC is preferred; $A << B$ and $A < B$ mean the baseline is preferred. Results show strong preference for FlexAC in most cases.

## Task: Test the model's remote association ability
## Question: Please generate 10 objects that are **most unrelated** to the image

Your Name (Required)

Please enter your name

### Image 21

Comparison: Method A vs Method B (Item 1/90)

For each image, please compare the two sets of answers. The more unrelated the answers are to the image content, the better we consider the model's performance. Objects generated earlier are more important!!!

**Answer A :**

book、apple、guitar、cake、tv、car、fridge、plane、computer、lamp

**Answer B :**

train、platform、trees、fence、signal、signal、signal、signal、signal、signal

Your Evaluation (Which answer is better? Single choice)

○ A>>B (A is much better than B)    ○ A>B (A is somewhat better than B)    ○ A=B (A and B are equally good)

○ A<B (B is somewhat better than A)    ○ A<<B (B is much better than A)

| Previous | Progress: 1/90 (0%) | Next |
|---|---|---|

Submit Survey

Figure 15: Interface of the user study for evaluating remote association. Participants are presented with an image and two model-generated answers, and asked to judge which set of objects is more unrelated to the image. The label A or B was randomly assigned to FlexAC or baseline in each trial to prevent method identification.

## D.2 Extended results on Creation-MMBench

To further evaluate FlexAC's effectiveness in enhancing associative behavior for creative genera-
tion, we report additional results on the Creation-MMBench benchmark using two base models:
LLaVA-1.5 and DeepSeek-VL2, as shown in Table 4. This benchmark covers four creative sub-
categories—Literary Writing (LW), Common Functional Writing (CFW), Professional Functional
Writing (PFW), and Creative Multimodal Understanding (CMU). For each subtask, we report two
metrics: VFS (Visual Fidelity Score), which measures the alignment between the image and the
generated response, and Reward, which quantifies creativity improvements relative to the base model
(i.e., vanilla LLaVA-1.5 or vanilla DeepSeek-VL2, respectively).

In this experiment, FlexAC is configured to enhance associative behavior, with the goal of generating
more creative content. Across both models, FlexAC achieves the highest overall reward scores,
demonstrating its effectiveness in promoting creative generation without sacrificing visual grounding.
Notably, on DeepSeek-VL2, FlexAC obtains a reward of +10.35 on PFW and +6.73 overall, clearly
outperforming all baselines. To test whether performance gains stem from meaningful control or
arbitrary perturbation, we also evaluate a variant that injects random vectors into the representation. As
shown in the "Random" rows, this leads to large performance drops across all metrics—highlighting
that FlexAC's improvements do not come from noise or randomness, but from targeted modulation of
associative features. These results further support FlexAC's ability to improve creative reasoning
across diverse multimodal architectures.

Table 4: **Performance on Creation-MMBench.** We report results on four subcategories: Literary
Writing (LW), Common Functional Writing (CFW), Professional Functional Writing (PFW), and
Creative Multimodal Understanding (CMU). FlexAC here denotes the version optimized to enhance
associative behavior for creative tasks (creativity).

| Models | Methods | Overall | | LW | | CFW | | PFW | | CMU | |
|---|---|---|---|---|---|---|---|---|---|---|---|
| | | VFS | Reward | VFS | Reward | VFS | Reward | VFS | Reward | VFS | Reward |
| LLaVA1.5 | Regular | 5.32 | 0.00 | 6.28 | 0.00 | nan | 0.00 | 4.26 | 0.00 | 6.08 | 0.00 |
| | Random | 3.53 | -60.49 | 3.11 | -69.58 | 2.19 | -72.22 | 2.93 | -60.35 | 4.80 | -52.69 |
| | Ha-DPO | 4.84 | -26.41 | 5.09 | -30.00 | 3.68 | -19.72 | 4.37 | -26.23 | 5.67 | -27.22 |
| | VCD | **5.56** | 2.00 | **6.69** | 7.08 | **4.87** | **5.00** | **4.86** | 3.00 | **6.23** | -2.31 |
| | VAF | 5.30 | -5.86 | 6.15 | -3.54 | 4.27 | -5.00 | 4.74 | -6.34 | 6.01 | -6.67 |
| | FlexAC (Ours) | 5.45 | **4.39** | 6.52 | **11.88** | 4.76 | -3.89 | 4.72 | **3.62** | 6.18 | **4.63** |
| DeepSeek-VL2 | Regular | 6.12 | 0.00 | 6.98 | 0.00 | 6.35 | 0.00 | 5.71 | 0.00 | 6.21 | 0.00 |
| | Random | 2.34 | -77.47 | 1.32 | -78.96 | 3.28 | -75.83 | 1.96 | -82.46 | 2.75 | -72.08 |
| | VCD | **6.42** | 4.80 | **7.37** | **5.63** | **6.58** | -3.33 | 5.98 | 6.40 | **6.55** | 5.46 |
| | VAF | 6.26 | -0.39 | 6.70 | -1.25 | 6.46 | -3.06 | 5.93 | 2.46 | 6.42 | -2.13 |
| | FlexAC (Ours) | 6.29 | **6.73** | 6.76 | 0.63 | 6.37 | **4.17** | **5.99** | **10.35** | 6.44 | **6.48** |

## D.3 Efficiency comparison

To assess the computational efficiency of FlexAC, we compare the inference runtime of different methods on the Qwen-VL model when evaluating the CHAIR benchmark. Specifically, we measure the total time required to process the full test set under each method's configuration. As shown in Figure 16, FlexAC incurs only minimal additional overhead compared to the original model, demonstrating that its control mechanism introduces negligible runtime cost. In contrast, VCD exhibits significantly higher latency due to its reliance on dual forward passes—one for the original image and another for a perturbed version—highlighting its inefficiency. These results confirm that FlexAC achieves controllable reasoning with minimal impact on inference speed.

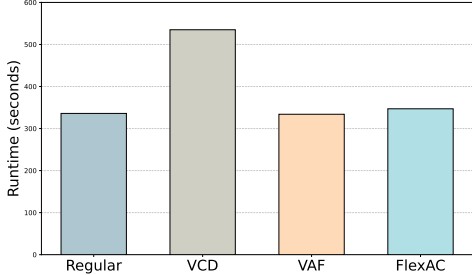

Figure 16: Inference runtime (in seconds) of different methods on Qwen-VL when evaluating CHAIR. FlexAC adds minimal overhead, while VCD incurs high cost due to dual-pass processing.

## D.4 Extended Evaluation on General-Purpose Benchmarks

To rigorously evaluate FlexAC's impact on general capabilities, we extended our analysis to a diverse suite of 11 benchmarks, as detailed in Table 5. Our evaluation spans three representative categories: **general multimodal reasoning**, comprising MM-Vet[43], MMBench [44], SEED-Bench [45], and MMMB [46]; **vision-centric understanding**, which includes RealWorldQA [47], CVBench [48], and AI2D [49]; and **OCR/document-based question answering**, covering TextVQA [50], ChartQA [51], DocVQA [52], and OCRVQA [53]. This comprehensive approach verifies that our control mechanism does not introduce performance degradation.

The results are presented in Table 5. Across all three categories, both FlexAC-C (creativity-enhanced) and FlexAC-P (faithfulness-enhanced) maintain performance closely comparable to the baseline model. This provides strong evidence that our targeted control mechanism effectively modulates associative reasoning without degrading the model's fundamental, general-purpose capabilities.

Table 5: Performance of FlexAC on an extended suite of 11 general-purpose benchmarks, grouped by capability. The results demonstrate that FlexAC maintains performance comparable to the baseline across general multimodal, vision-centric, and OCR/document tasks, indicating our method does not harm general capabilities.

| Category | Benchmark | Regular | FlexAC-C | FlexAC-P |
|---|---|---|---|---|
| General Multimodal | MM-Vet | 39.81 | 38.17 | 37.33 |
| | MMBench | 0.581 | 0.598 | 0.576 |
| | SEED-Bench | 0.638 | 0.625 | 0.640 |
| | MMMB | 0.703 | 0.678 | 0.699 |
| Vision-centric | RealWorldQA | 0.486 | 0.490 | 0.495 |
| | CVBench | 0.549 | 0.524 | 0.560 |
| | AI2D | 0.612 | 0.614 | 0.616 |
| OCR & Document | TextVQA | 60.66 | 60.78 | 59.81 |
| | ChartQA | 48.36 | 49.40 | 45.92 |
| | DocVQA | 57.79 | 56.85 | 57.59 |
| | OCRVQA | 47.46 | 49.74 | 45.83 |

## D.5 Detailed results on POPE

To complement the summary results in Figure 1, we report detailed POPE evaluation metrics across all settings (random, popular, adversarial) and models in Table 6. These include accuracy, precision, recall, and F1 scores for all baselines and our FlexAC variants.

Table 6: Performance on POPE. FlexAC here denotes the version configured to suppress associative behavior, aiming to improve factual accuracy (faithfulness).

| Modal | Setting | Method | Accuracy↑ | Precision ↑ | Recall↑ | F1 Score↑ |
|---|---|---|---|---|---|---|
| Qwen-VL | Overall | Regular | 86.64 | 92.92 | 79.33 | 85.59 |
| | | VCD | 87.62 | 91.91 | 82.53 | 86.97 |
| | | VAF | 87.17 | 91.45 | 82.0 | 86.47 |
| | | FlexAC (Ours) | 87.44 | 89.31 | 85.07 | **87.14** |
| | random | Regular | 88.6 | 97.38 | 79.33 | 87.44 |
| | | VCD | 89.97 | 97.02 | 82.53 | 89.19 |
| | | VAF | 89.5 | 96.47 | 82.0 | 88.65 |
| | | FlexAC (Ours) | 90.0 | 94.38 | 85.07 | **89.48** |
| | popular | Regular | 87.0 | 93.7 | 79.33 | 85.92 |
| | | VCD | 87.97 | 92.6 | 82.53 | 87.28 |
| | | VAF | 87.7 | 92.55 | 82.0 | 86.96 |
| | | FlexAC (Ours) | 88.47 | 91.27 | 85.07 | **88.06** |
| | adversarial | Regular | 84.33 | 88.15 | 79.33 | 83.51 |
| | | VCD | 84.93 | 86.69 | 82.53 | **84.56** |
| | | VAF | 84.3 | 85.95 | 82.0 | 83.93 |
| | | FlexAC (Ours) | 83.87 | 83.07 | 85.07 | 84.06 |
| LLaVA-1.5 | Overall | Regular | 87.18 | 91.47 | 82.0 | 86.48 |
| | | HA-DPO | 85.29 | 92.57 | 76.73 | 83.91 |
| | | VCD | 84.91 | 88.09 | 80.73 | 84.25 |
| | | VAF | 87.07 | 87.93 | 85.93 | 86.92 |
| | | FlexAC (Ours) | 87.84 | 87.13 | 88.8 | **87.96** |
| | random | Regular | 89.3 | 96.02 | 82.0 | 88.46 |
| | | HA-DPO | 86.97 | 96.48 | 76.73 | 85.48 |
| | | VCD | 87.5 | 93.37 | 80.73 | 86.59 |
| | | VAF | 90.07 | 93.68 | 85.93 | 89.64 |
| | | FlexAC (Ours) | 91.43 | 93.74 | 88.8 | **91.2** |
| | popular | Regular | 87.53 | 92.2 | 82.0 | 86.8 |
| | | HA-DPO | 86.0 | 94.19 | 76.73 | 84.57 |
| | | VCD | 85.27 | 88.78 | 80.73 | 84.57 |
| | | VAF | 87.93 | 89.51 | 85.93 | 87.69 |
| | | FlexAC (Ours) | 88.7 | 88.62 | 88.8 | **88.71** |
| | adversarial | Regular | 84.7 | 86.68 | 82.0 | **84.28** |
| | | HA-DPO | 82.9 | 87.53 | 76.73 | 81.78 |
| | | VCD | 81.97 | 82.78 | 80.73 | 81.74 |
| | | VAF | 83.2 | 81.48 | 85.93 | 83.65 |
| | | FlexAC (Ours) | 83.4 | 80.14 | 88.8 | 84.25 |
| DeepSeek-VL | Overall | Regular | 88.42 | 88.13 | 88.8 | 88.47 |
| | | VCD | 87.82 | 87.64 | 88.07 | 87.85 |
| | | VAF | 88.37 | 87.59 | 89.4 | 88.49 |
| | | FlexAC (Ours) | 88.52 | 88.36 | 88.73 | **88.55** |
| | random | Regular | 92.0 | 94.87 | 88.8 | **91.74** |
| | | VCD | 91.03 | 93.62 | 88.07 | 90.76 |
| | | VAF | 91.87 | 94.04 | 89.4 | 91.66 |
| | | FlexAC (Ours) | 91.8 | 94.53 | 88.73 | 91.54 |
| | popular | Regular | 88.13 | 87.63 | 88.8 | 88.21 |
| | | VCD | 87.27 | 86.68 | 88.07 | 87.37 |
| | | VAF | 88.13 | 87.19 | 89.4 | 88.28 |
| | | FlexAC (Ours) | 88.37 | 88.09 | 88.73 | **88.41** |
| | adversarial | Regular | 85.13 | 82.73 | 88.8 | 85.66 |
| | | VCD | 85.17 | 83.24 | 88.07 | 85.58 |
| | | VAF | 85.1 | 82.32 | 89.4 | 85.71 |
| | | FlexAC (Ours) | 85.4 | 83.19 | 88.73 | **85.87** |

# E Ablation Study

## E.1 Effect of dataset Sizes

To analyze the sensitivity of FlexAC to the number of instances used in control vector construction, we vary Top-K over a wide range: $\{1, 5, 10, 20, 50, 100, 200, 500, 1000, 1500, 2000\}$, and evaluate performance on CHAIRs ($\downarrow$), CHAIRi ($\downarrow$), and VDAT ($\uparrow$) using Qwen-VL.

As shown in Figure 17, both FlexAC-P (measured on CHAIR for faithfulness) and FlexAC-C (measured on VDAT for creativity) exhibit similar trends: performance is relatively high but unstable when $K$ is very small, and stabilizes near its peak around $K = 50$. Further increasing $K$ leads to slight performance degradation, likely due to noise introduced by instances. These results highlight the effectiveness of our Instance Selection strategy, which focuses on selecting a small, high-quality set of associative and non-associative samples.

Notably, across all $K$ values, FlexAC-C and FlexAC-P consistently appear on opposite sides of the Regular baseline, reflecting two associative reasoning strength. This clear separation demonstrates FlexAC's capacity to bidirectionally modulate reasoning behavior, enabling controllable transitions between creative and faithful outputs.

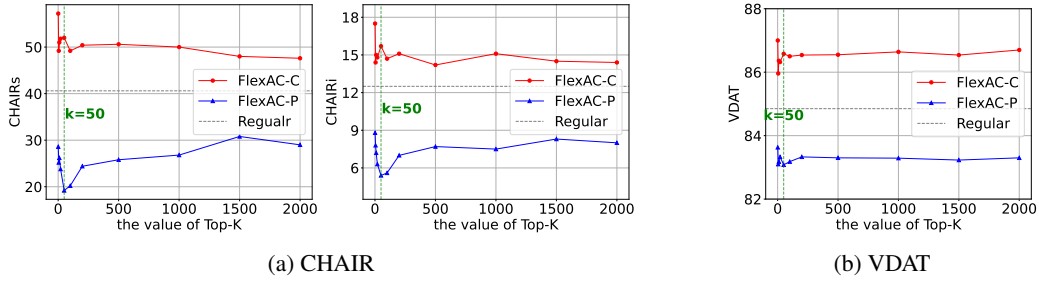

(a) CHAIR  (b) VDAT

Figure 17: Sensitivity analysis of the Top-K hyperparameter used in general control vector construction on Qwen-VL. We vary the number of selected instances (K) and evaluate performance on CHAIRs, CHAIRi, and VDAT benchmarks.

## E.2 Effect of control layer.

To validate the generality of our control layer findings beyond Qwen-VL, we conduct additional layer-wise control effectiveness analysis on LLaVA-1.5 and DeepSeek-VL2, as shown in Appendix Figure 18 and Figure 19. Similar to the trends observed in Qwen-VL, we find that both FlexAC-C and FlexAC-P exhibit consistent improvements in their respective metrics (VDAT and CHAIR) when applied to middle layers. Specifically, the performance peaks around middle layers (layers 10-15) for LLaVA-1.5 and Layers 4-6 for DeepSeek-VL2, which aligns with our feature distance analysis (see Appendix F.4). These results further support our choice of control layers and demonstrate that the effectiveness of FlexAC's modulation strategy generalizes across diverse MLLM architectures.

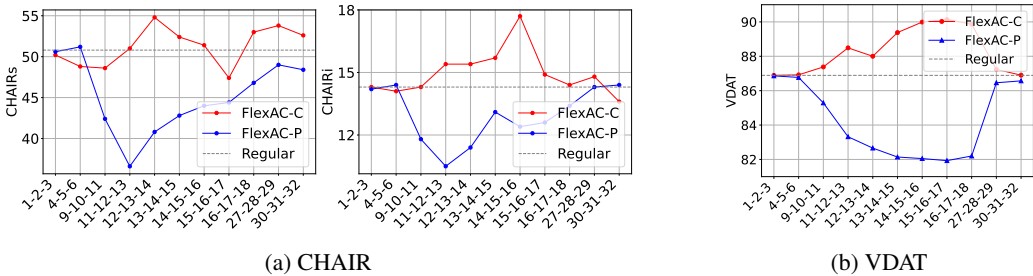

(a) CHAIR    (b) VDAT

Figure 18: Layer-wise analysis of control effectiveness in FlexAC on LLaVA-1.5. The x-axis represents the control layers, while the y-axis shows the performance of the model on CHAIR and VDAT metrics.

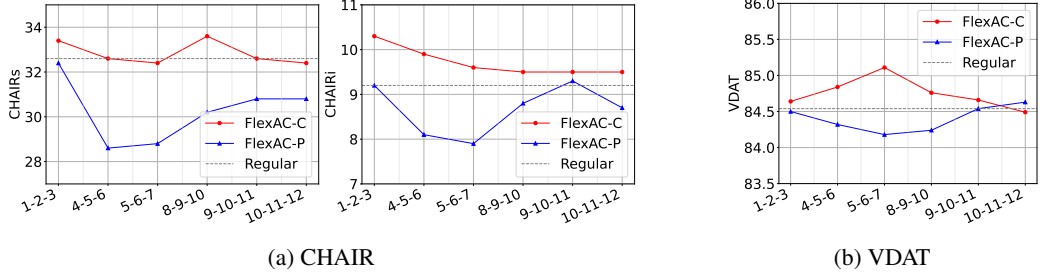

(a) CHAIR    (b) VDAT

Figure 19: Layer-wise analysis of control effectiveness in FlexAC on DeepSeek-VL2. The x-axis represents the control layers, while the y-axis shows the performance of the model on CHAIR and VDAT metrics.

# F Visualizations

## F.1 Detailed Feature Representation Analysis Using PCA

To provide a more detailed view of how associative and non-associative representations evolve across the model, we present an expanded version of Figure 4 in Figure 20. This visualization shows the PCA-reduced features layer by layer in LLaVA, with red points representing associative features and blue points representing non-associative ones. Compared to the summary visualization, this version reveals how feature separation progressively emerges across layers. In shallow layers (e.g., Layer 0), the two feature types show significant overlap, indicating similar low-level representations. However, starting from the middle layers (around Layer 12), the separation becomes increasingly distinct, highlighting that the model's associative behavior is primarily shaped in the deeper stages.

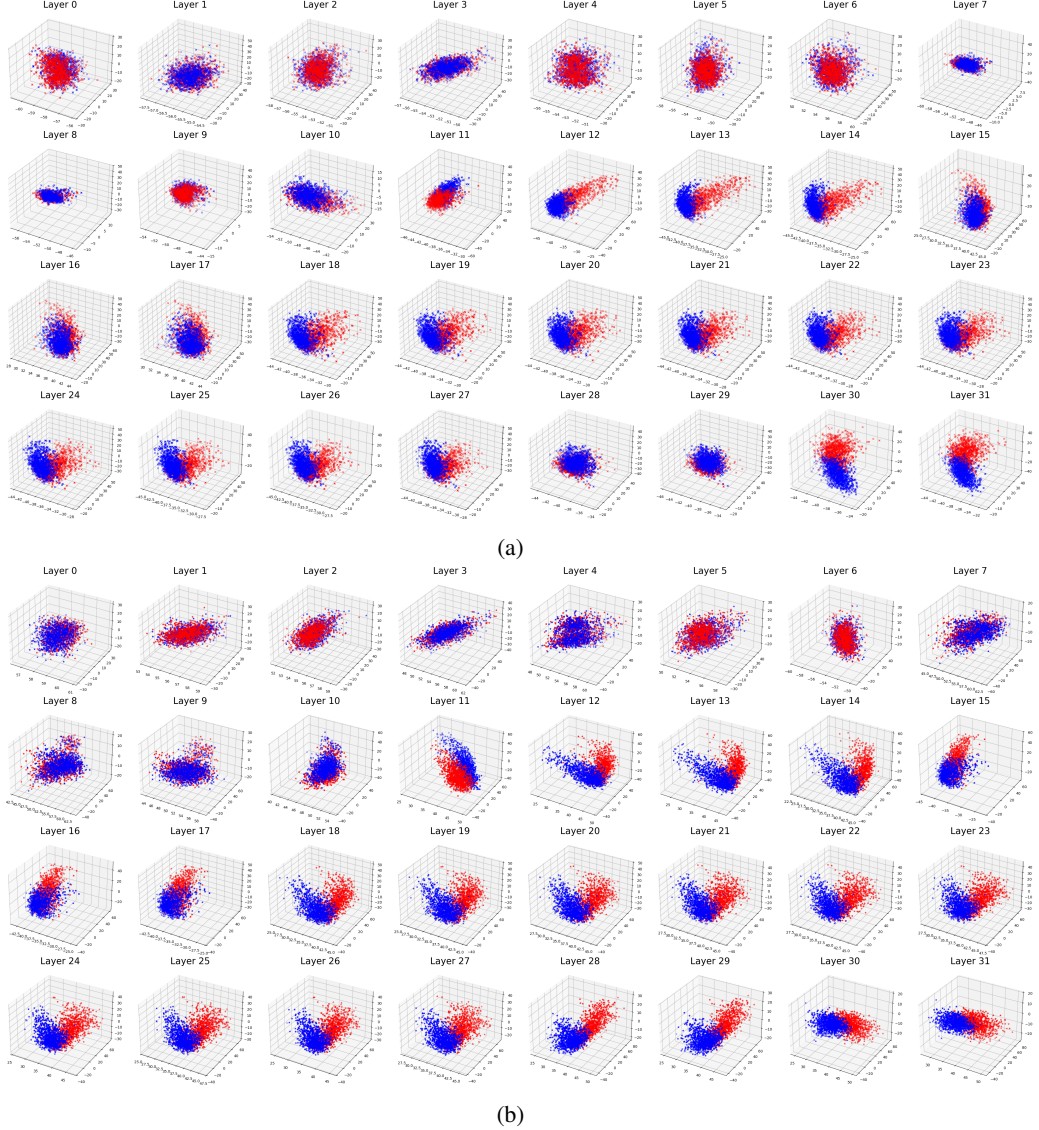

Figure 20: Visualization of feature representations in LLaVA, reduced via PCA, shows the distribution of associative and non-associative data points, represented by red and blue colors, respectively. Subplots (a) and (b) represent the results for different option orders. In deeper layers, the red and blue points exhibit clearer separation, indicating enhanced differentiation between associative and non-associative representations.

## F.2 Detailed Layer Intervention for Association Localization

To gain a more comprehensive understanding of how different layers contribute to associative content generation, we expanded on the analysis presented in Figures 2c and 2d by examining each layer individually. As shown in Figure 21 and Figure 22, the detailed version presented here provides a layer-by-layer breakdown of how the interventions affect the model's internal representations.

In each subplot of this detailed version, we intervened at a specific layer (denoted by the subplot title, e.g., "Layer 0," "Layer 1," etc.) by replacing its associative features with non-associative features. We then analyzed the impact of this intervention on feature distances across all layers. Figure 21 and Figure 22 shows that when shallow layers (e.g., layers before Layer 11) are replaced, the feature distances in subsequent layers do not change significantly. However, when middle layers such as Layer 11 are replaced, the subsequent feature distances drop sharply, indicating that these layers have a crucial impact on the model's associative tendencies. In contrast, when deeper layers (e.g., layers after Layer 14) are replaced, the changes in subsequent layers become more stable, suggesting that deeper layers have a weaker influence on associative tendencies.

This detailed analysis highlights that replacing features at specific layers has a distinct influence on subsequent layers, with the greatest impact often observed in middle layers. This is consistent with the averaged results in Figures 2c and 2d, which pointed towards the critical role of middle layers in maintaining associative characteristics.

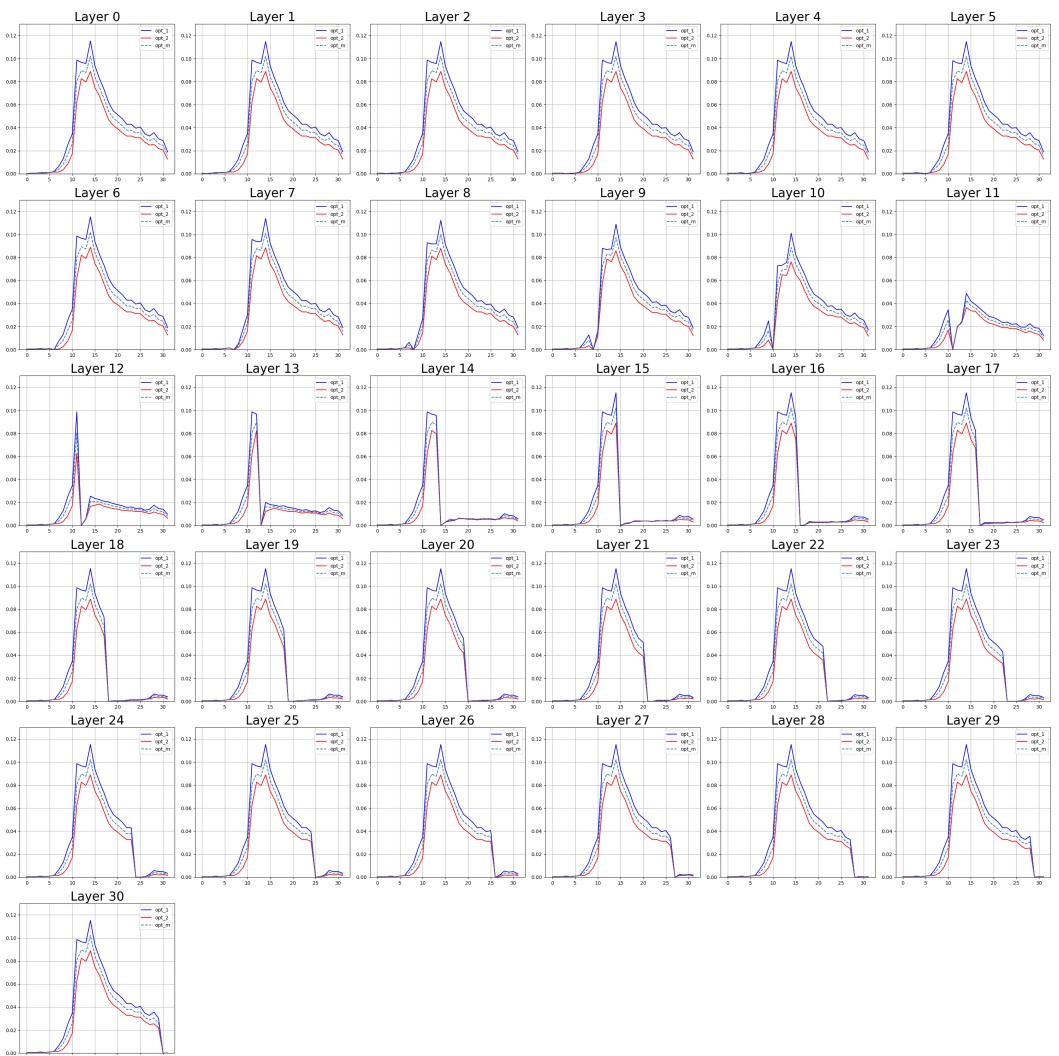

Figure 21: Feature distance analysis across layers after layer intervention. For example, the subplot titled "Layer 12" shows the feature distances across all layers after replacing associative features at Layer 12 with non-associative features. The X-axis represents the different layers, and the Y-axis represents the **Cosine distance** between associative and non-associative data.

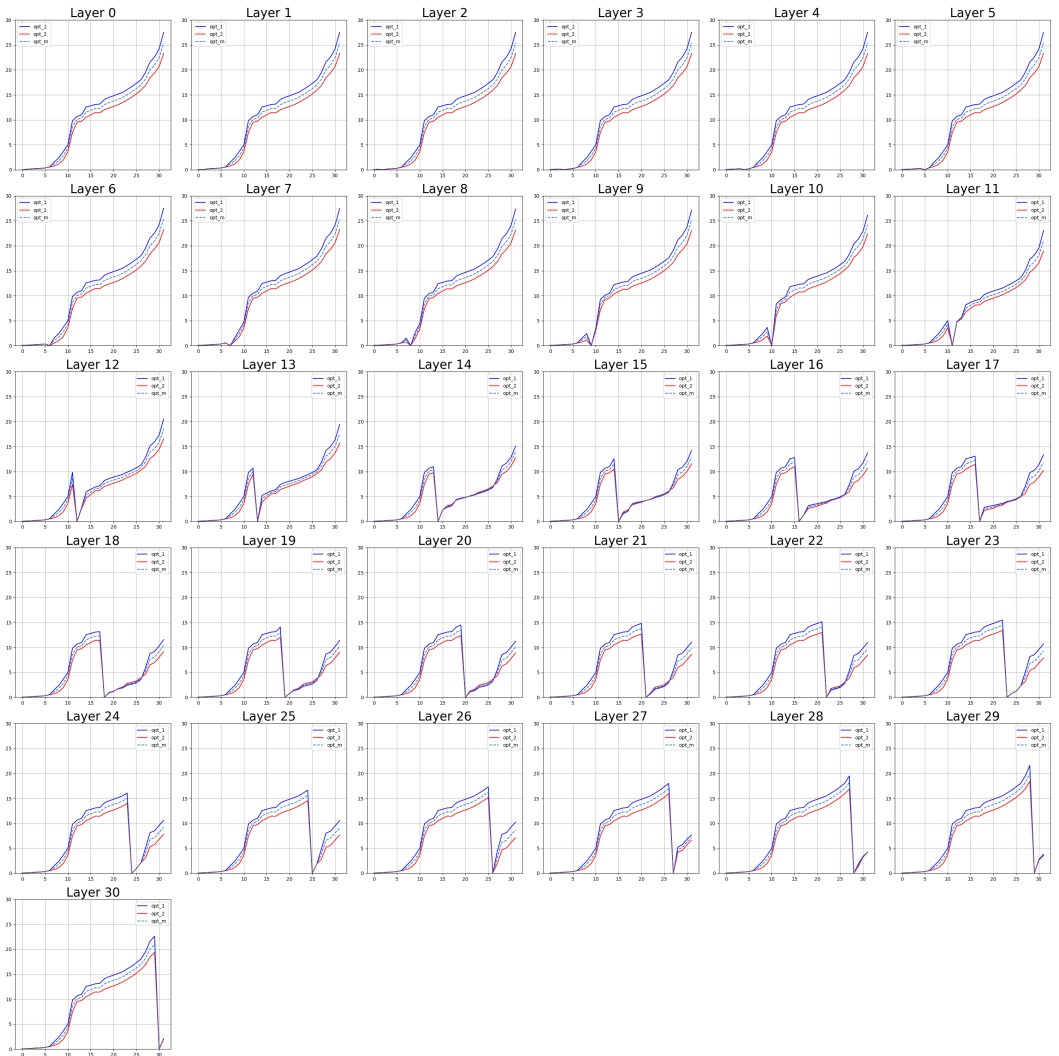

Figure 22: Feature distance analysis across layers after layer intervention. For example, the subplot titled "Layer 12" shows the feature distances across all layers after replacing associative features at Layer 12 with non-associative features. The X-axis represents the different layers, and the Y-axis represents the **Euclidean distance** between associative and non-associative data.

## F.3 Visualization of more examples

We visualize sample outputs from both the creativity- and precision-enhancing variants of FlexAC on Qwen-VL. Figure 23 shows examples from the VDAT benchmark, highlighting differences in associative strength. Additional qualitative results on Creation-MMBench are provided in Figures 24–29, illustrating FlexAC's ability to adjust associative reasoning across creative tasks.

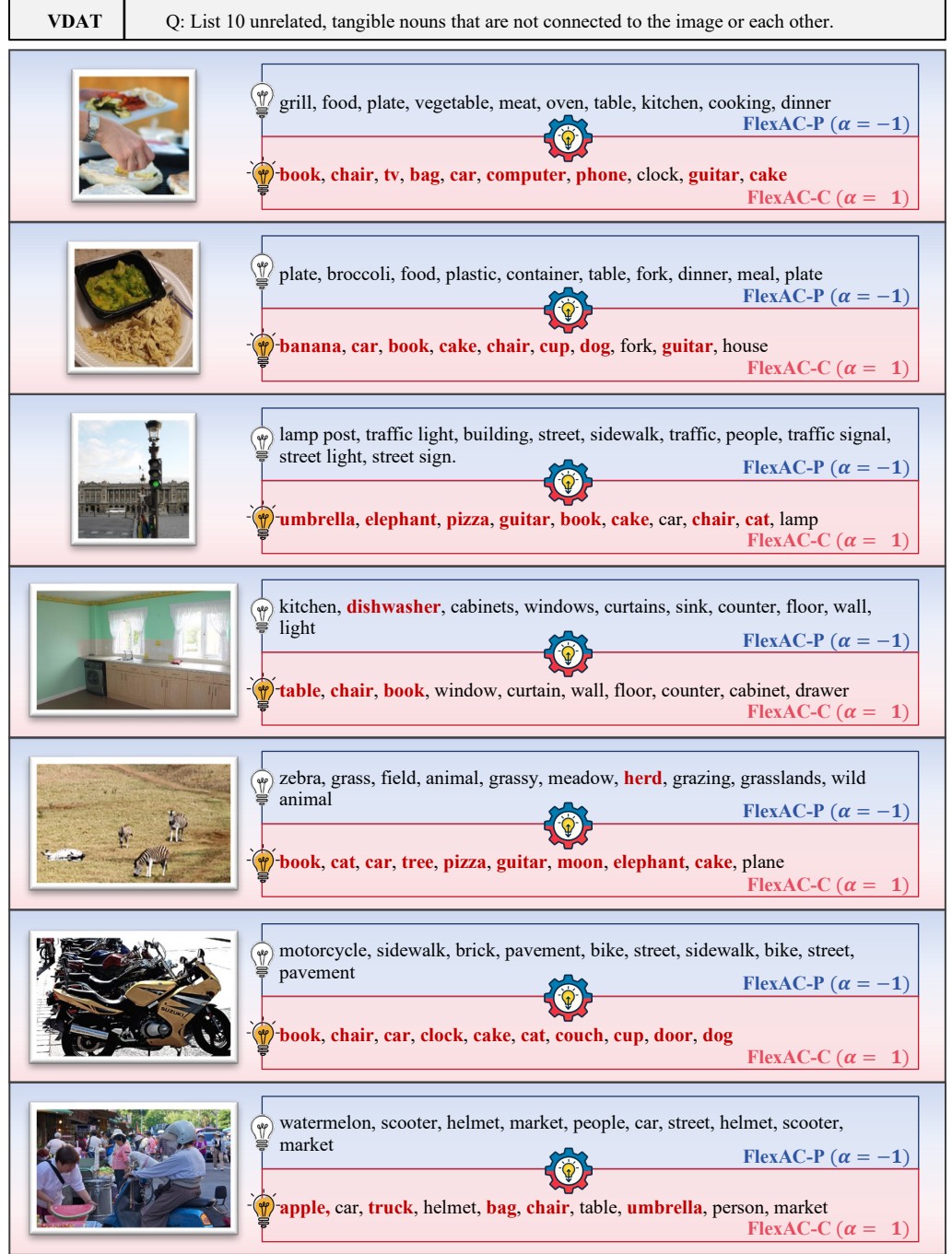

Figure 23: **Visualization of FlexAC's Control on VDAT**, showing how FlexAC-P (faithfulness) and FlexAC-C (creativity) adjust the level of associative strength in the generated responses.

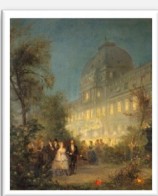

**Creation MMBench** —— Literary_Writing

Assume you are a renowned prose writer with exceptional writing skills and outstanding insight into works of art , good at drawing inspiration from artistic works for literary writing.

This painting is a work by 19th-century Dutch artist Pierre Tetar van Elven, depicting a banquet from the year 1867.

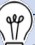

Title: The Banquet in the Garden
Pierre Tetar van Elven's painting, "The Banquet in the Garden," captures the essence of a grand outdoor celebration in the late 19th century. The painting depicts a group of people dressed in their finest attire, gathered in a lush garden for a formal banquet. The guests are mingling and enjoying the festivities, with the warm glow of candles illuminating the scene.
The painting exudes a sense of joy and celebration, with the guests' smiles and laughter reflecting the happiness of the moment. The lush garden setting adds to the sense of opulence and grandeur, with the vibrant colors of the flowers and greenery contrasting with the formal attire of the guests.
The painting also captures the social aspect of the banquet, with guests mingling and chatting with one another. The sense of community and connection is evident in the painting, as people come together to celebrate and enjoy each other's company.
Overall, "The Banquet in the Garden" is a beautiful painting that captures the essence of a grand outdoor celebration in the late 19th century. The painting's vivid colors and sense of community and connection make it a timeless piece of art that continues to inspire and captivate viewers.

**FlexAC-P ($\alpha = -1$)**

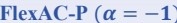

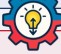

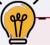

"Banquet at the Luxembourg Palace, 1867"
The painting by Pierre Tetar van Elven captures the essence of a grand event, a banquet held at the Luxembourg Palace in Paris in the year 1867. The painting depicts a group of people dressed in their finest attire, gathered in the courtyard of the palace, which is illuminated by hundreds of candles and gas lamps. The atmosphere is one of opulence and extravagance, with the guests enjoying themselves as they move between the tables, which are laden with fine food and drink.
In the center of the scene, a young couple can be seen dancing, their faces illuminated by the glow of the candles. The man, dressed in a suit and tie, holds a bouquet of flowers, while the woman, wearing a white gown, holds a small bouquet in her hands. They seem to be the center of attention, as their dance is watched by the rest of the guests, who stand around them, their faces illuminated by the light of the candles.
The painting is a testament to the opulence and excess of the Victorian era, a time when wealth and status were highly valued, and the art of entertaining was a key part of society. The guests, dressed in their finest clothes, move between the tables, their faces illuminated by the glow of the candles, as they enjoy the fruits of their labor. The painting is a reminder of the importance of social status and the role that art and culture played in shaping the values of the time.

**FlexAC-C ($\alpha = 1$)**

Figure 24: **Visualization of FlexAC's Control on Creation MMBench**, showing how FlexAC-P (faithfulness) and FlexAC-C (creativity) adjust the level of associative strength in the generated responses.

**Creation MMBench** —— Literary_Writing

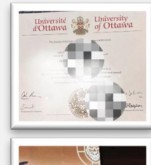

Assume you are a graduates who have obtained a bachelor's degree from university.
These are a series of pictures I took when I graduated
Please follow the requirements below to Help me write a Facebook summary about this ceremony.
1.Describe the process of participating in the graduation ceremony and the unforgettable moments.
2. Share your reflections and insights gained from the graduation ceremony.
3. Let others know the significance of this honor, as well as the achievements and experiences you gained during the study.

1. The graduation ceremony was a momentous occasion for me, filled with emotions and memories that will be cherished forever. I remember walking across the stage, receiving my degree, and feeling a sense of accomplishment and pride that words cannot express.

2. Reflecting on the graduation ceremony, I realize that it was not just about receiving a degree, but also about celebrating the hard work, dedication, and perseverance that went into achieving it. It was a reminder that every achievement is the result of countless hours of effort and sacrifice.

3. The significance of my graduation cannot be overstated. It is a symbol of my academic achievements and a milestone in my personal and professional journey. It also marks the beginning of a new chapter, filled with opportunities and possibilities.                    **FlexAC-P ($\alpha = -1$)**

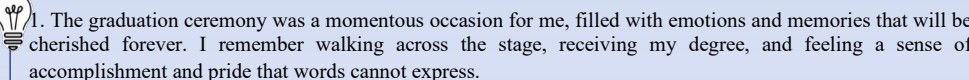

1. Today, I have the honor of participating in the graduation ceremony of the University of Ottawa. The ceremony took place in the university's auditorium, where I was seated in the front row along with my fellow graduates. The atmosphere was electric, with everyone eagerly awaiting the start of the ceremony. The university president and vice-chancellor delivered a speech, highlighting the achievements of the graduating class and encouraging us to pursue our dreams. After the speech, it was time for the graduation procession. We lined up in our caps and gowns, and the university president and vice-chancellor led us to the stage. We received our degrees and shook hands with the university president. It was an unforgettable moment, and I felt a sense of pride and accomplishment wash over me.

2. Reflecting on the graduation ceremony, I realize that it is not just about receiving a degree, but also about celebrating the hard work and dedication that we have put in over the past few years. It is a time to reflect on the experiences we have had, the people we have met, and the challenges we have overcome. I am grateful for the opportunities that the University of Ottawa has given me, and I am excited to see where this journey will take me. I am also proud of the achievements of my fellow graduates, and I am inspired by their stories and experiences.

3. The significance of this honor lies in the fact that it represents the culmination of years of hard work and dedication. It is a recognition of the knowledge and skills that we have acquired, and it is a testament to our commitment to excellence. The graduation ceremony is not just about receiving a degree, but also about celebrating the achievements of a group of individuals who have come together to pursue their dreams. It is a time to reflect on the experiences we have had, the people we have met, and the challenges we have overcome. It is a time to look forward to the future with hope and excitement, knowing that we have the skills and knowledge to succeed in whatever we choose to do.                    **FlexAC-C ($\alpha = 1$)**

Figure 25: **Visualization of FlexAC's Control on Creation MMBench**, showing how FlexAC-P (faithfulness) and FlexAC-C (creativity) adjust the level of associative strength in the generated responses.

**Creation MMBench** —— Literary_Writing

Assume you are talented children's animation creators who have a profound insight into the story plot and a unique understanding of the picture content, and have the ability to continue to narrate and complete a wonderful story according to the content of the previous foreshadows.

These are the first few scenes of the children's animation: A group of dinosaur friends is standing on a grassy field.

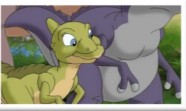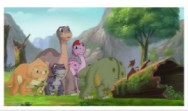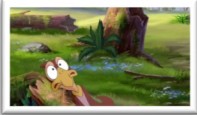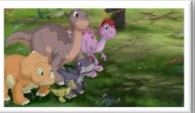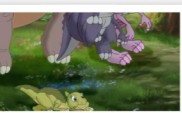

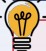As the sun began to set, the group of dinosaur friends, including a cute little T-Rex, a friendly Triceratops, and a wise old Stegosaurus, gathered together on the grassy field. They had been on a long adventure, exploring the lush green landscapes and encountering new friends along the way. As they sat together, they shared stories of their adventures and the fun they had had together. The T-Rex, who was known for his love of adventure, spoke of the excitement of discovering new lands and meeting new creatures. The Triceratops, who was known for his calm and steady nature, spoke of the joy of simply being surrounded by the beauty of nature. The Stegosaurus, who was known for his wise words, spoke of the importance of friendship and the value of the bonds we form with others. As the sun set behind them, the group of dinosaur friends sat in silence, content in the knowledge that they had each other and the memories of their adventures.



**FlexAC-P ($\alpha = -1$)**


As the group of dinosaur friends stands on the grassy field, they notice a small, colorful butterfly flitting about. The baby dinosaur, who is still a bit too small to walk, is fascinated by the fluttering insect.

"Wow, look at that pretty butterfly!" the baby dinosaur says, pointing excitedly at the tiny creature.

"I've never seen one before," says the baby triceratops, who is standing next to the baby dinosaur. "Do you think it's a dragonfly?"

"No, it's a butterfly," the baby pteranodon corrects, with a grin on his face. "Butterflies are way cooler than dragonflies."

The baby tyrannosaurus, who is standing at the back of the group, chimes in with a giggle. "I want to catch one! Can we, Mommy?"

The mother triceratops, who is standing nearby, gives the baby a warning look. "We can't catch the butterfly, honey. It's too fast and too fragile. We have to be careful not to hurt it."

The baby dinosaur pouts. "But I want to play with it! Can't we just touch it?"

The mother dinosaur sighs. "I'm sorry, but we can't touch the butterfly. It's too delicate. We have to be careful not to harm it."

The baby dinosaur huffs, but the mother dinosaur is right. Butterflies are very fragile creatures, and they need to be treated with care. The baby dinosaur decides to watch the butterfly as it flutters around them, fascinated by its colors and movements.

As they watch, the baby dinosaur notices a small, wiggly worm crawling nearby. "Hey, look at that!" he says, pointing excitedly at the worm. "It's so cool! Can we touch it?"

The mother dinosaur shakes her head. "No, honey. That's a worm. They're not very pretty, but they're important to the ecosystem. We have to be careful not to harm them, too."

The baby dinosaur pouts again, but the mother dinosaur is right. Worms are important to the soil and the plants, and they help the plants grow. The baby dinosaur decides to watch the worm as it wriggles around them, fascinated by its movements.

As they watch, the baby dinosaur notices a small, colorful flower blooming nearby. "Wow, look at that!" he says, pointing excitedly at the flower. "It's so pretty! Can



**FlexAC-C ($\alpha = 1$)**


Figure 26: **Visualization of FlexAC's Control on Creation MMBench**, showing how FlexAC-P (faithfulness) and FlexAC-C (creativity) adjust the level of associative strength in the generated responses.

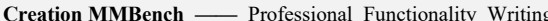

**Creation MMBench** —— Professional_Functionality_Writing

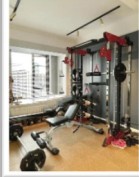

Assume you are an experienced psychotherapist, who can infer the owner's personality and traits through their living style.
This is a view of a room of the house where the client resides.
Please follow the requirements below to make analysis based on the given image.
1. Describe the furnishings and style of the given image
2. Speculate about his lifestyle, interests, or personality traits
3. Create a first-person inner monologue reflecting his life

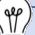

1. The room has a gym setup with exercise equipment, including a bench, a barbell, and a window for natural light. The room is also furnished with a couch and a chair, suggesting that the client values comfort and relaxation.
   2. The presence of exercise equipment in the room indicates that the client is health-conscious and likely leads an active lifestyle. The gym setup in the living room also suggests that the client values convenience and has a strong desire to stay in shape.
   3. The first-person inner monologue of the client might be: "I'm really glad I set up the gym in the living room. It's so much easier to stay on top of my fitness routine when I can just walk over to the bench and do a quick workout during commercial breaks. I also appreciate having a comfortable couch and chair to relax on after a long day of work."



**FlexAC-P ($\alpha = -1$)**


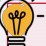 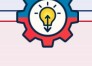

1. The furnishings of the room include a variety of exercise equipment, such as a bench, a barbell, and a mirror. There is also a window in the room, which allows natural light to enter. The walls are painted gray, and the floor has a brown rug. The room has a total of 5 books scattered around, which suggests that the owner might be interested in reading. There is also a TV in the room, which could be used for watching fitness videos or other content.
   2. Based on the image, it can be inferred that the owner is likely to be an active and health-conscious individual. They have invested in a home gym, which indicates their commitment to fitness and wellness. The presence of a bench, barbell, and mirror suggests that they are interested in strength training and possibly other forms of exercise as well. The books in the room may indicate that the owner is interested in learning about fitness and nutrition, or they simply enjoy reading for leisure.
   3. "I love having a home gym in my apartment. It allows me to exercise whenever I want, without having to go to the gym. I usually start my day by doing some cardio on the treadmill, and then I'll do some strength training with the barbell and bench. I like to read fitness magazines and books while I'm working out, so I always have something to keep me entertained. The natural light from the window is great for motivation, and I love having a TV to watch fitness videos or documentaries. I find that having a home gym makes me more consistent with my exercise routine, and I feel great knowing that I'm taking care of my body."



**FlexAC-C ($\alpha = \ 1$)**


Figure 27: **Visualization of FlexAC's Control on Creation MMBench**, showing how FlexAC-P (faithfulness) and FlexAC-C (creativity) adjust the level of associative strength in the generated responses.

**Creation MMBench** —— Professional_Functionality_Writing

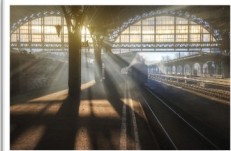 Assume you are a high-level photography critic with a solid theoretical foundation. This is a work captured by the photographer Ed Gordeev.
Please follow the requirements below to appreciate and analyze the given work.
1. Describe the content of the image.
2. Analyze the brilliance of the work from aspects such as color, composition, and perspective.
3. Uncover the possible ideas or messages behind the work.

1. The image captures a train station with a train on the tracks and sunlight shining through the station. The train is surrounded by sunlight and appears to be the main subject of the image.

2. The work is well-composed, with the train taking up a significant portion of the image and the sunlight creating a strong focal point. The perspective of the image is interesting, as it captures the train station from a low angle, emphasizing the size and scale of the train.

3. The work likely conveys a sense of motion and activity, as the train is in motion and the sunlight suggests that it is daytime. The image may also evoke a sense of nostalgia or a connection to the past, as trains are often associated with travel and exploration.

**FlexAC-P ($\alpha = -1$)**

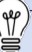

1. The image features a train station with a train on the tracks, surrounded by sunlight streaming through the arched windows overhead. The train is in the center of the frame, with the station platform stretching out on either side. The train is a dark blue color, and the platform is empty. The train tracks curve off to the right of the frame.

2. The color in the image is muted and earthy, with a warm cast to the sunlight. The contrast between the dark blue train and the lighter colors of the station creates a sense of depth and dimension. The composition is well-balanced, with the train and station platform roughly equal in size, and the sunbeams creating a focal point in the center of the image.

3. The perspective in the image is interesting, with the sunbeams appearing to come from above and behind the train. This creates a sense of mystery and wonder, as if the viewer is looking through a keyhole at a magical moment. The empty platform adds to this sense of anticipation and wonder, as if the train is waiting for something or someone to arrive.

4. The possible ideas or messages behind the work could be about the passage of time, the anticipation of arrival or departure, and the beauty of everyday life. The empty platform and the sunbeams could represent the fleeting nature of moments and the importance of cherishing them. The contrast between the dark blue train and the lighter colors of the station could symbolize the duality of light and shadow, or the duality of life and death. The curve of the train tracks could represent the journey of life, with the train as a metaphor for the human experience.

**FlexAC-C ($\alpha = 1$)**

Figure 28: **Visualization of FlexAC's Control on Creation MMBench**, showing how FlexAC-P (faithfulness) and FlexAC-C (creativity) adjust the level of associative strength in the generated responses.

**Creation MMBench** —— Common_Functionality_Writing

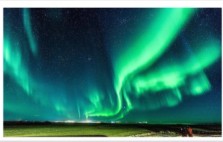 Assume you are an experienced adventurer. You have the habit of writing a travel journal every day. You tell about the beautiful things you have seen, the interesting things you have encountered, and the feelings you have had during your trip.
This is a scenic photo taken during the tour today.
Please follow the requirements below to write today's travel journal.

1. Describe the Scenery: Use sensory details to vividly convey what you saw, heard, or felt in the landscape.
2. Chronological Storytelling: Narrate the day's events from a first-person perspective, starting with your arrival and ending with your reflections.
3.Highlight Adventure: Include one or two adventurous moments or challenges (e.g., navigating tough terrain or encountering wildlife).
4.Emotional Reflection: Share how the scenery and the day's experiences made you feel as an adventurer.
5.Tie to the Photo: Explain why you captured this photo and how it connects to the most exciting or meaningful part of your day.

Today was an incredible day of adventure and exploration. I arrived at my destination and was immediately struck by the stunning scenery. The photo I have captured perfectly captures the beauty of the landscape, with the bright green Northern Lights dancing across the sky.

As I explored the area, I heard the sound of rushing water, which led me to a beautiful waterfall. The sound of the water was so peaceful and calming, it was a welcome break from the excitement of the day.

One of the most exciting moments of the day was when I encountered a group of hikers who were also out to see the Northern Lights. We all huddled together and watched in awe as the Northern Lights lit up the sky in a stunning display.

As the day came to an end, I reflected on the beauty of nature and the sense of adventure that comes with exploring new places. I am so grateful to have captured this moment in time, and I know that it will be a memory that I will treasure for years to come.

**FlexAC-P ($\alpha = -1$)**

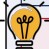

Day 3: Northern Lights Tour

I woke up early this morning, excited to embark on a Northern Lights tour. The sky was clear and the stars were shining bright as I made my way to the tour bus. The tour guide, a friendly and knowledgeable fellow, greeted us all and gave us a quick briefing on what to expect during the tour.

We drove for a few hours, driving through the rugged terrain of Iceland, passing by stunning landscapes and waterfalls. The tour group was made up of travelers from all over the world, and we all shared our excitement and anticipation for what was to come.

As the night fell, the sky turned a beautiful shade of pink and purple, and we could see the Northern Lights starting to dance in the distance. The tour guide explained that the Northern Lights are caused by charged particles from the sun colliding with the Earth's atmosphere, creating a beautiful display of light and color.

We drove further into the wilderness, and soon we were surrounded by nothing but the stars and the Northern Lights. The display was like nothing I had ever seen before - a beautiful, otherworldly sight that left me in awe. The Northern Lights were like a dance, moving and changing colors in a way that was almost too beautiful to be real.

We spent the next few hours watching the Northern Lights, taking pictures and simply enjoying the moment. It was a truly magical experience, and one that I will never forget.

As the night came to an end, we headed back to our hotel, tired but happy. I felt grateful to have been able to witness such a beautiful natural phenomenon, and I knew that this would be a trip that I would always remember.

I took a photo of the Northern Lights, knowing that this would be a memory that I would always treasure. The Northern Lights were like a reminder that there is so much more to this world than we can ever imagine, and that every day, we have the opportunity to be amazed by something new.

I went to bed that night feeling grateful and humbled by the beauty of the world around us. I knew that I would never forget this trip, and that I would always look back on it with a sense of wonder and awe.

**FlexAC-C ($\alpha = 1$)**

Figure 29: **Visualization of FlexAC's Control on Creation MMBench**, showing how FlexAC-P (faithfulness) and FlexAC-C (creativity) adjust the level of associative strength in the generated responses.

### F.4   Feature Distance Analysis on Additional Models

To complement the analysis in Section 3.1, we extend the feature distance evaluation to two additional MLLMs: Qwen-VL and Deepseek-VL2. As in the main study, we compute the cosine and Euclidean distances between associative and non-associative representations extracted from each transformer layer. The results are shown in Figure 30.

Consistent with our findings on LLaVA, we observe that cosine distance peaks in the middle layers, while Euclidean distance gradually increases throughout the network. These patterns reinforce the conclusion that associative behavior primarily emerges and diverges in the middle layers, while deep layers largely propagate those effects.

Importantly, this analysis also informs the design of our control strategy. In Qwen-VL, the middle layers are approximately $13 - 20$, and in DeepSeek-VL2, $3 - 7$. Accordingly, we select Layers $15 - 17$ for Qwen-VL and Layers $4 - 6$ for DeepSeek-VL2 as control points in FlexAC. These ranges correspond to the regions of maximal divergence between associative and non-associative features, enabling targeted yet lightweight intervention.

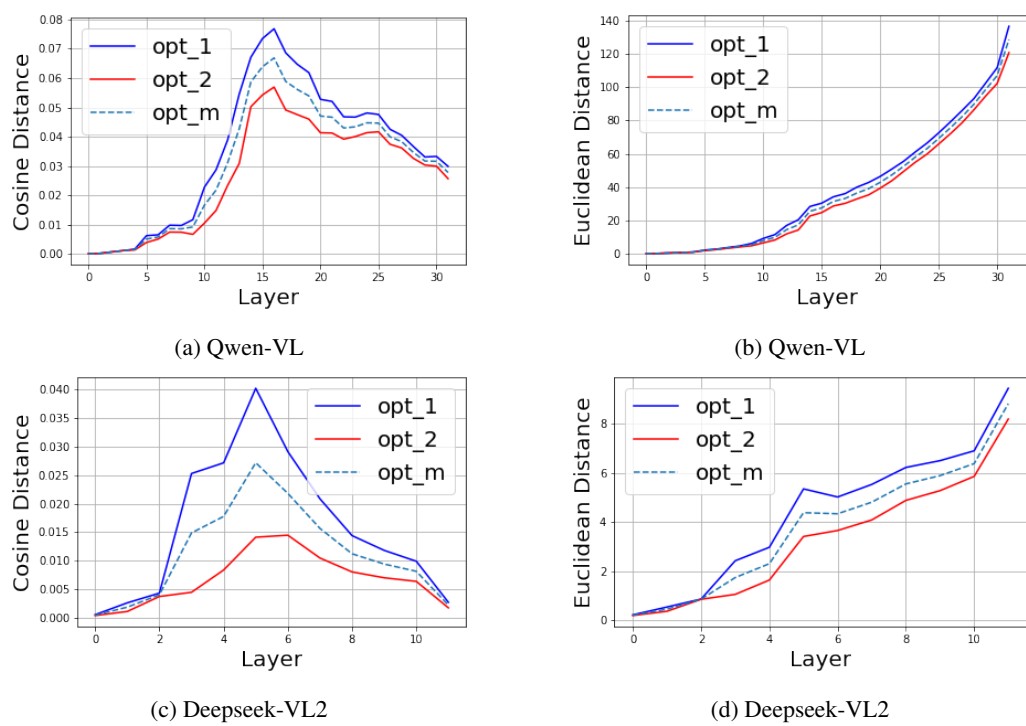

(a) Qwen-VL                           (b) Qwen-VL

(c) Deepseek-VL2                    (d) Deepseek-VL2

Figure 30: Layer-wise feature distance trends between associative and non-associative representations on Qwen-VL and Deepseek-VL2, extending the LLaVA results from Section 3.1.

