# OpenReview forum: "FlexAC: Towards Flexible Control of Associative Reasoning in Multimodal Large Language Models"
_NeurIPS.cc/2025/Conference — NeurIPS 2025 poster_

### Official Review · Reviewer_Up2p · 2025-06-24

**Clarity:** 2
**Significance:** 2
**Originality:** 2
**Rating:** 4
**Confidence:** 4

**Summary:**

The paper presents FlexAC, a lightweight method for steering associative reasoning in multimodal LLMs by modifying middle-layer model parameters. Using parameter vectors extracted from hallucinated outputs, the approach adjusts the model’s balance between factual accuracy and creativity. Experiment results validate the steerability on hallucination and creativity.

**Questions:**

1. Since the non-middle layers show minimal change on hallucination, have the authors explored applying steering vectors to all layers as an ablation?

2. Does FlexAC operate solely on the language-side layers, or is the vision encoder also considered? A brief discussion would clarify the scope of parameter mixing.

3. Equations (7) and (9) suggest both unnormalized and normalized vector mixing. Given the presence of LayerNorm in the transformer, can the authors comment on whether there are redundant computations involved?

4. While most prior efforts on steerable LLMs focus on trainable approaches, they are still relevant and should be discussed in the related work section. Including these works would help position FlexAC more clearly within the broader landscape of controllable and steerable LLM/MLLMs.

**Ethical Concerns:**

["NO or VERY MINOR ethics concerns only"]

**Final Justification:**

I thank the authors for their effective rebuttal, which has clarified my main questions regarding the method's robustness and generalization.

While my major concerns have been resolved, I feel the connection drawn between the 'hallucination vector' and 'associative reasoning' could be presented with more nuance. I believe that further clarifying the assumptions behind this conceptual link would be a valuable addition to the final paper.

In light of the authors' successful efforts in addressing my primary concerns, I have increased my score from 3 to 4.

**Limitations:**

Yes.

**Quality:**

3

**Strengths And Weaknesses:**

**Strengths**

1. The identification of middle layers as critical for hallucination control is novel and well-supported by empirical analysis. The proposed approach for steering between faithfulness and creativity is thoughtfully designed.

2. The analysis of associative behavior is well-motivated and convincingly executed, especially the layer-wise representation study. These analyses provide solid empirical grounding for the design of the control mechanism.

3. The introduction of the VDAT benchmark fills an evaluation gap for associative reasoning in MLLMs.



**Weaknesses**

1. **Data quality**: The construction of the steering vector depends on hallucinated input-response pairs, which are derived via blurred inputs and specific prompting. This may introduce distributional biases, e.g., capturing only specific hallucination types. Besides that, top-k selection can be sensitive to outlier data. Some robustness or sensitivity analysis of the vector construction would strengthen the approach.

2. **Connection between hallucination vector and associative reasoning**: While the extracted steering vector clearly controls the factual grounding, it's less certain whether it directly reflects a model’s associative reasoning capability. The paper seems to assume that hallucinated features correspond to associative reasoning, but this isn't explicitly validated. For instance, in Figure 9 (b)–(c), FlexAC-P (presumably more powerful on associative reasoning) performs similarly or worse than FlexAC-C on reasoning-heavy tasks like MMMU and Math. Therefore, it seems to me this method is for hallucination control instead of associative reasoning control.

3. **Model Architecture**: The empirical analysis on layer-wise impact on associative reasoning are demonstrated only on LLaVA model. It remains unclear whether the same behavior holds for other architectures, particularly mixture-of-experts (MoE) models.

---

> ### Author Rebuttal · Authors · 2025-07-31
>
> _**Weakness 1. Data quality: The construction of the steering vector depends on hallucinated input-response pairs, which are derived via blurred inputs and specific prompting. This may introduce distributional biases, e.g., capturing only specific hallucination types. Besides that, top-k selection can be sensitive to outlier data. Some robustness or sensitivity analysis of the vector construction would strengthen the approach.**_
>
> **Responses:**
> - **Hallucination Distribution**: We analyzed our data and confirmed it contains a rich mixture of both **object-level (86.35%) and relation-level (94.8%) hallucinations**, ensuring it is not biased toward a specific type.
> - **Sensitivity Analysis of Top-K**: Our method demonstrates robustness in two key aspects: the `top-k` selection is not sensitive to outliers, and the optimal `k` value is generalizable across datasets.
> 	- **Generalizability of `k=50`:** As shown in the table I and our original appendix (Figure. 17), `k=50` is a robust choice that delivers near-peak performance across all tested benchmarks (CHAIR, VDAT, MME, HallusionBench), making it a reliable and practical default setting.
> 	- **Robustness to Outliers:** The method is not sensitive to including a large number of instances (and potential outliers). For instance, on MME (Perception), performance degrades gracefully from the peak at `k=50` (1427.8) to `k=2000` (1390.0), rather than collapsing. This confirms the robustness of our vector construction.
>
> Table I: Sensitivity analysis of Top-K selection.
>
> |Benchmark|k=1|k=50|k=100|k=1000|k=2000|
> |-|-|-|-|-|-|
> |MME(Perception)|1397.3|1427.8|1423.7|1397.4|1390.0|
> |MME(reasoning)|371.8|413.6|413.8|406.1|405.7|
> |HallusionBench|50.6|50.6|50.3|49.1|49.2|
>
> ---
> _**Weakness 2. Connection between hallucination vector and associative reasoning: While the extracted steering vector clearly controls the factual grounding, it's less certain whether it directly reflects a model’s associative reasoning capability. The paper seems to assume that hallucinated features correspond to associative reasoning, but this isn't explicitly validated. For instance, in Figure 9 (b)–(c), FlexAC-P (presumably more powerful on associative reasoning) performs similarly or worse than FlexAC-C on reasoning-heavy tasks like MMMU and Math. Therefore, it seems to me this method is for hallucination control instead of associative reasoning control.**_
>
> **Responses:**  We believe there may be a critical misunderstanding of our notation which, once clarified, resolves the core concern.
>
> - **A Crucial Clarification: The Reviewer's Premise on P vs. C is Inverted.**
> The reviewer’s analysis mistakenly assumes that `FlexAC-P` enhances associative reasoning, which contradicts our definition. In fact, the correct premise is as follows: **`FlexAC-C` (Creativity, α = 1)** is designed to **ENHANCE** associative reasoning; **`FlexAC-P` (Precision, α=-1)** is designed to **SUPPRESS** it.
>
> - **Re-interpreting Figure 9 with the Correct Premise：** With the correct premise, the data in Figure 9 now strongly supports our claim. It shows that `FlexAC-C` (the creativity-enhanced version) maintains stable performance on logical reasoning tasks (MMMU/Math). This is a key feature: **our method precisely targets _associative reasoning_ without disrupting the model's separate, core _logical reasoning_ capabilities.**
> - **Direct Validation on Creativity Tasks.** The fact that our vector controls more than just hallucination is validated by `FlexAC-C`'s strong performance on creativity-focused benchmarks: It generates abstract, metaphorical content on **Creation-MMBench**. It improves divergent thinking scores on **VDAT**, a result confirmed by our **human study (Appendix, Sec. 4.1)**.
>
> ---
> _**Weakness 3. Model Architecture: The empirical analysis on layer-wise impact on associative reasoning are demonstrated only on LLaVA model. It remains unclear whether the same behavior holds for other architectures, particularly mixture-of-experts (MoE) models.**_
>
> **Responses:** Our submitted appendix (App. 6.4, Fig. 30) already provides feature analysis for Qwen-VL and **the MoE-based DeepSeek-VL2**. To provide even more direct evidence, the following intervention analysis for DeepSeek-VL2 shows each layer's impact; **a larger '$\Delta$' value signifies a greater influence on the final result:**
>
> |Intervention Layer|Last Layer Euclidean Distance|$\Delta$ (↓)|
> |:-:|:-:|:-:|
> |0|8.8|–|
> |1|8.6|-0.2|
> |2|8.6|-0.0|
> |3|8.1|-0.5|
> |**4**|**6.8**|**-1.3**|
> |**5**|**5.5**|**-1.3**|
> |**6**|**4.1**|**-1.4**|
> |7|2.9|-1.2|
> |8|2.3|-0.6|
> |9|1.7|-0.6|
> |10|1.4|-0.3|
> |11|1.2|-0.2|
>
> The table's "$\Delta$" column peaks at Layer 6 (-1.4), confirming our paper's finding that the middle layers (4-7) are the critical control region. This result from the MoE model is consistent with our main analysis. The full plots will be added to the camera-ready version.
>
> ---
> _**Question 1. Since the non-middle layers show minimal change on hallucination, have the authors explored applying steering vectors to all layers as an ablation?**_
>
> **Responses:** Per your advice, we conducted an ablation study applying the steering vector to all layers.
>
> Table II: Comparison of Middle-Layer vs. All-Layer Intervention in LLaVA.
>
> | Methods               | CHAIRs | CHAIRi | Recall | Len  |
> | --------------------- | ------ | ------ | ------ | ---- |
> | Regular               | 50.8   | 14.3   | 79.7   | 97.3 |
> | FlexAC (middle-layer) | 36.6   | 10.5   | 75     | 95.1 |
> | FlexAC (all-layers)   | 0.2    | 0.6    | 7.1    | 22.3 |
>
> While the 'all-layers' intervention drastically reduces CHAIR scores, it comes at the cost of catastrophic damage to the model's generative ability. This is evidenced by the collapse in performance (e.g., LLaVA's Recall dropped from 75.0 to 7.1) and the frequent generation of meaningless outputs like _empty strings or gibberish_. This result confirms that all-layer intervention is harmful, thus validating our design choice to precisely target only the middle layers. The possible reason is that all-layer intervention disrupts both low-level representations and high-level semantics, leading to unstable decoding and degraded output.
>
> ---
> _**Question 2. Does FlexAC operate solely on the language-side layers, or is the vision encoder also considered? A brief discussion would clarify the scope of parameter mixing**_
>
> **Responses:**  FlexAC operates **solely on the language-side layers**, a deliberate choice based on the standard MLLM architecture.
>
> In MLLMs, the vision encoder functions as the perceptual "eyes," while the LLM is the cognitive "brain" responsible for reasoning[1]. Since hallucination and creativity are **cognitive phenomena** that occur in the "brain," our intervention is precisely targeted at the LLM's layers where this reasoning takes place
>
> [1] A survey on multimodal large language models (National Science Review 2024)
>
> ---
> _**Question 3. Equations (7) and (9) suggest both unnormalized and normalized vector mixing. Given the presence of LayerNorm in the transformer, can the authors comment on whether there are redundant computations involved?**_
>
> **Responses:** **No redundant computation is involved.** Our normalization (Eq. 9) and the transformer's LayerNorm serve distinct, complementary purposes:
> - **Our Eq. 9 (Norm Preservation):**
> 	- **Target:** A **single** hidden state vector after we have steered it.
> 	- **Purpose:** To reset the steered vector's magnitude to match the **original** vector's magnitude. This is a stability measure ensuring our intervention only changes the representation's _direction_, not its strength. This is a common practice in activation steering[1,2,3].
> - **Transformer's LayerNorm:**
> 	- **Target:** The **entire set** of all token vectors in the sequence.
> 	- **Purpose:** To rescale the _distribution_ of this entire group of vectors (e.g., to zero mean, unit variance) to ensure a stable input for the **next** network layer.
>
> [1] Reducing hallucinations in large vision-language models via latent space steering (ICLR 2025)
>
> [2] Steering away from harm: An adaptive approach to defending vision language model against jailbreaks(CVPR 2025)
>
> [3] Safety arithmetic: A framework for test-time safety alignment of language models by steering parameters and activations(EMNLP 2024)
>
> ---
> _**Question 4. While most prior efforts on steerable LLMs focus on trainable approaches, they are still relevant and should be discussed in the related work section. Including these works would help position FlexAC more clearly within the broader landscape of controllable and steerable LLM/MLLMs.**_
>
> **Responses:** We agree and will update the "Related Work" section to better position FlexAC relative to trainable steering methods. The discussion will include:
> - **Preference-Based Alignment:** Methods like Reinforcement Learning from Human Feedback (RLHF)[1] and Direct Preference Optimization (DPO)[2], which fine-tune the model on preference data.
> - **Targeted Fine-tuning:** Approaches like CogSteer[3], which fine-tunes lightweight adapter modules in specific middle layers to steer behavior.
> - **Auxiliary Module Training:** Frameworks that train a separate module to guide the LLM, such as SCAV[4] (trains a linear classifier to find a steering vector) or TSV[5] (trains a lightweight vector to be added during inference).
>
> [1] Training language models to follow instructions with human feedback (NeurIPS 2022)
>
> [2] Direct preference optimization: Your language model is secretly a reward model (NeurIPS 2023)
>
> [3] Cogsteer: Cognition-inspired selective layer intervention for efficiently steering large language models. (ACL Finding 2024)
>
> [4] Uncovering Safety Risks of Large Language Models through Concept Activation Vector (NeurIPS 2024)
>
> [5] Steer LLM Latents for Hallucination Detection (ICML 2025)

---

> > ### Author Response · Authors · 2025-08-03
> >
> > Dear Reviewer,
> >
> > We hope our responses have adequately addressed your previous concerns. We look forward to hearing from you and would be happy to address any remaining concerns that you may still have.
> >
> > Thanks,
> >
> > The Authors

---

> ### Comment · Reviewer_Up2p · 2025-08-05
>
> **W1 (Data quality):** Thank you for the clarification; this has resolved my initial concern. The empirical analysis on the sensitivity of top-k is convincing and demonstrates the robustness of your results for k>1. I suggest you incorporate the specific breakdown of hallucinations ('object-level: 86.35%' and 'relation-level: 94.8%') into the final paper, as it adds valuable detail.
>
> **W2 (FlexAC-C vs. FlexAC-P Performance):** I appreciate the clarification on the distinction between FlexAC-C and FlexAC-P. However, the question remains regarding the performance divergence. Given that FlexAC-C maintains stable performance on logical reasoning tasks like MMMU and Math, why does it underperform on the MMStar logical reasoning benchmark? Furthermore, the performance comparison between P v.s. C in the "Science & Technology" and "Instance Reasoning" categories are also notable. *Could you provide a systematic discussion on these discrepancies between the 'C' and 'P' variants as shown in Figure 9?*
>
> Furthermore, my concern about the conceptual link between the 'hallucination vector' and 'associative reasoning' persists. 'Associative reasoning' is a broad cognitive concept that is difficult to formalize, and I am not convinced that it can be directly equated with the identified hallucination vector. The paper would be stronger if this claim were either better substantiated or framed more cautiously.
>
> **W3 (MoE architecture analysis):** The additional analysis on MoE is clear and addresses my questions. Thank you.
>
> **Q1 (Generative Ability):** The observation that intervention across all layers leads to catastrophic damage to the model's generative ability is an important finding. I recommend including this analysis in the final version of the paper
>
> **Q2 & Q3 & Q4:** Thank you for the clarifications; my questions on these points are resolved.

---

> ### Author Response · Authors · 2025-08-06
> **1. Systematic Discussion on the Performance of 'C' and 'P' Variants on MMStar (Part 1/2)**
>
> Thank you for your thoughtful review and for acknowledging that our previous rebuttal has resolved your concerns regarding data quality (W1), MoE architecture analysis (W3), and your other questions. We are greatly encouraged by your positive feedback.
>
> We would like to offer further clarification on the two insightful final points you raised.
>
> ### 1. Systematic Discussion on the Performance of 'C' and 'P' Variants on MMStar
>
> We appreciate this detailed question, as it allows us to present a more systematic discussion. As shown in Figure 9, the performance divergence between FlexAC-C and FlexAC-P is an **insightful and, indeed, an expected outcome of our work**.  The performance of each variant is directly tied to how well its reasoning style aligns with the demands of a specific task.
>
> - **1.1 For tasks requiring highly focused, convergent thinking, FlexAC-P performs better than FlexAC-C**:
>     What these tasks have in common is that they typically have a single, objective correct answer that must be reached through rigorous, step-by-step reasoning. The model must suppress irrelevant associations and focus strictly on the given visual evidence.
>
>     - **Tasks:** 'Logical reasoning', 'Instance reasoning', 'Science & technology'.
>     - **Why does FlexAC-P perform better?**: FlexAC-P suppresses association to enhance focus. This allows the model to act like a meticulous analyst, strictly following instructions and logical chains. It concentrates on the most direct evidence at hand, excelling in tasks that demand precision and objectivity.
>     - **Why does FlexAC-C perform worse?**: FlexAC-C's strength lies in activating a broad network of associations, which becomes a form of "cognitive noise" here. It can introduce irrelevant, distracting ideas that cause the model to deviate from the correct logical path and make flawed judgments based on these unrelated associations.
>
>    - **Specific Task Examples:**
>         - **Instance reasoning:**
>           - **Task Requirement:** Precisely compare specific attributes of two or more objects (e.g., quantity, color, size).
>           - **Scenario:** The question is, "Is the number of yellow squares in Image A greater than in Image B?"
>           - **How FlexAC-P (Precision Mode) succeeds:** It strictly executes the instructions: 1) Locate all squares in Image A; 2) Filter for the yellow ones; 3) Count them. It repeats this process for Image B and then compares the numbers. Its attention is locked onto the core attributes of "yellow" and "square."
>           - **Why FlexAC-C (Creativity Mode) fails:** When seeing the yellow squares in Image A, it might associate them with "cheese" or "sunshine." When viewing the squares in Image B, their arrangement might make it think of "buildings" or "Tetris." These irrelevant associations interfere with the core task of counting, leading to miscounts or a flawed comparison.
>         - **Logical Reasoning**:
>           - **Task Requirement:** Follow step-by-step reasoning based on a chart, code, or a set of visual rules.
>           - **Scenario:** A flowchart instructs: "If the shape is a circle, follow the red arrow; if it's a square, follow the blue arrow."
>           - **How FlexAC-P (Precision Mode) succeeds:** It meticulously executes the logical judgment: 1) Identify the current shape as a "circle." 2) Look up the rule corresponding to "circle." 3) Strictly follow the instruction to "follow the red arrow."
>           - **Why FlexAC-C (Creativity Mode) fails:** It is prone to "over-interpretation." Seeing a red-colored circle, it might incorrectly force an association between the object's color and the "red arrow," even though the rule only specifies shape. Alternatively, it might observe the overall layout of the flowchart and guess the path based on visual "harmony" or "symmetry" rather than the explicit rule, causing the reasoning to break down.
>
> 	  - **Special Case: (Science & technology)** Performance is on par here because the task requires both precise visual observation (benefiting from P-mode's focus) and the flexible connection of phenomena to abstract scientific principles (a divergent skill). Therefore, the 'P' and 'C' variants have similar performance.

---

> ### Author Response · Authors · 2025-08-06
> **1. Systematic Discussion on the Performance of 'C' and 'P' Variants on MMStar (Part 2/2)**
>
> - **1.2 For tasks that benefit from flexible matching and creative problem-solving, FlexAC-C performs better than FlexAC-P.**:The solution paths for these tasks are often not fixed and may require connecting complex visual patterns to abstract concepts or discovering non-obvious "shortcuts."
>     - **Tasks:**  'Coarse/Fine-grained Perception', 'Math'.
>     - **Why does FlexAC-C perform better?**: FlexAC-C enhances associative abilities to promote "divergent thinking." This allows the model to think outside the box, connecting seemingly unrelated visual elements to form a meaningful whole or devising novel approaches to solve a problem.
>     - **Why does FlexAC-P perform worse?**: FlexAC-P's high focus becomes a form of "cognitive rigidity" in this context. It may over-focus on the literal, local details of an image while failing to grasp its holistic, abstract meaning. It can also get stuck on standard solutions, unable to adapt flexibly.
>
>    - **Specific Task Examples:**
>         - **'Coarse/Fine-grained Perception'**:
>           - **Task Requirement:** Identify an object's abstract category (e.g., car brand, painting style) based on visual cues.
>           - **Scenario:** Judge a car's brand from a blurry close-up photo of only its front grille.
>           - **How FlexAC-C (Creativity Mode) succeeds:** It can creatively match multiple weak cues—the blurry shape of the grille, the faint outline of its pattern, the reflection of light—with a wide net of associations in its knowledge base, like "BMW's kidney grille" or "Audi's single-frame grille." It synthesizes these weak signals into a correct identification even with incomplete information.
>           - **Why FlexAC-P (PrecisionMode) fails:**  It over-focuses on the image's physical details, asking, "How many pixels are black?" or "What is the exact curvature of this line?" Because the image is blurry, these precise details are useless. By suppressing association, it cannot make the conceptual leap from these imperfect local features to the abstract concept of a brand, likely concluding "unknown" due to the lack of a precise match.
>         - **'Math'**:
>           - **Task Requirement:** Solve visual math problems, especially geometry tasks that require a "flash of insight."
>           - **Scenario:** Calculate the area of an irregular shape in a geometry problem.
>           - **How FlexAC-C (Creativity Mode) succeeds:** Through association, it might recall strategies of "cutting" and "rearranging." It could creatively try adding an "auxiliary line" to the diagram, decomposing the irregular shape into a simple rectangle and a triangle. This act of adding a line is a non-linear, creative problem-solving step.
>           - **Why FlexAC-P (Precision Mode) fails:** It would see the irregular shape and try to directly apply the standard area formulas it knows (for rectangles, circles, etc.). Finding that none apply, it gets stuck. Its focused nature makes it difficult to perform "destructive" modifications on the original image (like adding a line) because it prefers to analyze the given visual information as-is, thereby missing the key to the solution.
>
> In summary, the performance divergence on MMStar is not a defect but a demonstration of the fundamental trade-off between creative, divergent thinking and precise, convergent reasoning. **We will add this detailed analysis to the final manuscript.**

---

> > ### Author Response · Authors · 2025-08-06
> > **2. On the Conceptual Link Between 'Hallucination Vector' and 'Associative Reasoning'**
> >
> > ### 2. On the Conceptual Link Between 'Hallucination Vector' and 'Associative Reasoning'
> >
> > We agree completely with your assessment that "associative reasoning" is a broad and difficult-to-formalize cognitive concept. To make this concept computationally tractable, our work employs **a necessary simplification**, which we support with the following three-part argument:
> >
> > - **2.1 Theoretical Grounding and Operationalization:** Our work is inspired by cognitive science theories of **convergent** and **divergent thinking** [1]. We frame faithfulness as low-association strength (convergent) and creativity as high-association strength (divergent). Within this spectrum, hallucination is modeled as an extreme, uncontrolled form of divergent thinking. This is further supported by neuroscience research suggesting a mechanistic link between creativity and hallucination [2]. Therefore, using the vector extracted from hallucinated states serves as a practical way to represent and control the strength of these associations.
> >
> > - **2.2 Empirical Validation:**
> >
> >     - **Direct Control over Associative Strength:** Critically, our results in Figure 3 validate that this vector can steer a behavioral spectrum. By adjusting a single coefficient, α, we can simultaneously guide both the hallucination metric (CHAIR score: 38.8 → 53.6) and the associative reasoning metric (VDAT score: 83 → 87.9). This shows a tight empirical linkage between our hallucination-derived vector and a validated measure of associative strength.
> >
> >     - **Impact on Creative Tasks:** This controlled associative strength translates directly to functional creative performance. As our results show, the creativity-enhanced FlexAC-C variant achieves the highest scores on our VDAT benchmark. This enhanced associative ability corresponds to a state-of-the-art creativity "Reward" score of 10.92 on the Creation-MMBench [3], outperforming all baselines while maintaining high visual fidelity. This consistency validates that our "hallucination vector" effectively guides a core mechanism of associative reasoning crucial for creativity.
> >
> > - **2.3 Mitigating Multi-Dimensionality with Task-Specific Vectors:** We fully agree with the reviewer that a single vector cannot capture the full breadth of associative reasoning. Our framework is explicitly designed to mitigate this by moving beyond a simple 'hallucination vector'.  To handle tasks requiring different associative patterns, FlexAC augments the general associative vector with task-specific ones, enabling better adaptation to diverse creative tasks.
> >
> > Therefore, a more precise conclusion is that our work does not equate the 'hallucination vector' with the entire concept of 'associative reasoning.' Instead, we identify and isolate a **'hallucination-guided, precisely controllable vector' that serves as an effective operational proxy** for enhancing or suppressing the associative processes relevant to creativity. **Per your invaluable suggestion, we will revise the manuscript to more rigorously frame this conceptual model and the multifaceted nature of our approach.**
> >
> > Thank you once again for your constructive and detailed feedback, which has significantly helped improve the rigor of our work.
> >
> > [1] The standard definition of creativity (Creativity research journal 2012)
> >
> > [2] To create or to recall? Neural mechanisms underlying the generation of creative new ideas (NeuroImage 2014)
> >
> > [3] Creation-MMBench: Assessing Context-Aware Creative Intelligence in MLLM (ICCV 2025)

---

> > > ### Comment · Reviewer_Up2p · 2025-08-06
> > >
> > > I thank the authors for their thorough and systematic rebuttal.
> > > - The detailed discussion comparing FlexAC-P and FlexAC-C for the Figure 9 results was particularly insightful. I encourage the authors to incorporate this analysis into the final manuscript.
> > > - I also appreciate the clarification on the conceptual link between the 'hallucination vector' and 'associative reasoning'. I strongly encourage authors to revise the paper by (1) adding more discussions on the conceptual link and theories from cognitive science; and (2) explicitly stating the simplifying assumptions in the paper.
> > >
> > > Considering that the authors' rebuttal addresses my concerns, I raise my score.

---

> > > > ### Author Response · Authors · 2025-08-07
> > > >
> > > > We sincerely thank the reviewer for their positive feedback and for engaging so thoughtfully with our rebuttal. We are delighted that our clarifications have addressed their concerns and are grateful for their decision to raise their score.
> > > >
> > > >
> > > >
> > > > We appreciate the specific, constructive suggestions for improving the manuscript and will incorporate them into the final version as follows:
> > > >
> > > >
> > > >
> > > > - **Discussion of FlexAC-P vs. FlexAC-C:** As recommended, we will integrate the detailed discussion comparing FlexAC-P and FlexAC-C directly into the manuscript. We agree that this analysis, which was previously in the rebuttal, is valuable for readers to fully understand the results related to Figure 9.
> > > >
> > > >
> > > >
> > > > - **Conceptual Framework and Assumptions:** We will act on the strong encouragement to revise the conceptual framing. The manuscript will be updated to:
> > > >
> > > >     - 1. Include a more thorough discussion on the conceptual link between the 'hallucination vector' and 'associative reasoning', drawing connections to established theories in cognitive science to ground our work.
> > > >
> > > >     - 2. Explicitly state and clarify the simplifying assumptions underlying our model and methodology. We agree this is critical for transparency and helps frame the scope and limitations of our contribution.
> > > >
> > > >
> > > >
> > > > We are confident that these revisions will significantly strengthen the paper. We thank the reviewer again for their invaluable guidance.

---

### Official Review · Reviewer_2JXU · 2025-06-28

**Clarity:** 2
**Significance:** 2
**Originality:** 2
**Rating:** 4
**Confidence:** 4

**Summary:**

This paper proposes FlexAC (Flexible Association Control), a training-free framework for modulating associative reasoning strength in multimodal large language models (MLLMs). The authors investigate how associative behavior emerges in MLLMs and find that middle layers are critical for shaping associative tendencies. Based on these findings, they develop a method that uses hallucination-guided steering vectors to control the trade-off between faithfulness and creativity. The approach is evaluated on hallucination benchmarks (CHAIR, POPE), creativity benchmarks (VDAT, Creation-MMBench), and general-purpose benchmarks (MME, MMMU, MMStar).

**Questions:**

1. The steering vector is defined as the difference between hallucinated and grounded representations. Could this vector be capturing "non-factualness" or "context-ignorance" rather than genuine creativity? Have you explored alternative methods for deriving creativity-enhancing vectors using high-quality human-generated creative content instead of model-induced hallucinations?
2. How does VDAT performance correlate with functional, grounded creative tasks? Is there evidence that pushing VDAT scores higher doesn't simply train the model to ignore visual input, undermining its utility as a multimodal model?
3. The method requires empirical identification of optimal control layers for each model. How sensitive is performance to layer selection? For practitioners with new MLLMs, is there a principled way to identify these layers beyond exhaustive testing?
4. How well does the method transfer across different types of creative tasks beyond the evaluated benchmarks? What evidence supports that the observed behavior represents genuine creativity rather than reduced factual grounding?

**Ethical Concerns:**

["NO or VERY MINOR ethics concerns only"]

**Final Justification:**

Overall, the additional explanations and evidence provided in the rebuttal have addressed several of the original concerns. Although some aspects could still be refined, the progress is clear and encouraging. I raised the score.

**Limitations:**

yes

**Quality:**

2

**Strengths And Weaknesses:**

Strengths:
- The paper provides an interesting unified view linking hallucination and creativity through associative reasoning mechanisms, which offers a fresh theoretical perspective on MLLM behavior.
- The layer-wise analysis identifying middle layers as critical for associative behavior is well-conducted and provides valuable insights into MLLM internal mechanisms.
- The proposed method requires no additional training, making it practically appealing and computationally efficient.

Weaknesses:
- The central premise that creativity and hallucination represent two ends of a single "associative strength" spectrum is a significant oversimplification.
- The core technique of manipulating model behavior through activation steering is well-established. This work is essentially a direct application of existing representation steering concepts to MLLMs.
- A significant portion of creativity evaluation depends on VDAT, a new benchmark that measures ability to generate nouns unrelated to an image. This is an extremely narrow and potentially misleading proxy for divergent thinking. A model could score highly on VDAT by simply ignoring the image, which contradicts the goal of image-grounded creativity.

---

> ### Author Rebuttal · Authors · 2025-07-31
>
> _**Weakness 1. Associative Strength Spectrum Is Oversimplified.**_
>
> **Responses:** We agree this is a simplification, but argue it's an **effective operational model** rather than an oversimplification.
> - **Theoretical Grounding**:  Inspired by neuroscience research linking creativity and hallucination[1], we propose an 'associative strength' spectrum to model this connection. Drawing on theories of **convergent and divergent thinking**[2], we frame faithfulness as low associative strength (convergent) and creativity as high (divergent), with hallucination seen as an extreme form of uncontrolled divergence
> - **Empirical Validation**: Critically, Fig. 3 validates this model. By adjusting a single coefficient, α, we make it success to control both hallucination (CHAIR: 38.8→53.6) and associative strength (VDAT: 83→87.9). This provides strong empirical evidence that the Associative Strength Spectrum is an effective and practical abstraction for steering between creativity and hallucination
> - **Beyond a Single Spectrum**: Our framework goes **beyond** a unidimensional two-end simplification. We acknowledge the existence of **diverse creative trajectories between creativity and hallucination**. To capture this complexity, we introduce task-specific associative vectors that guide the model along multiple creative pathways, thereby addressing the inherently multidimensional nature of creativity
>
> [1] To create or to recall? Neural mechanisms underlying the generation of creative new ideas(NeuroImage 2014)
>
> [2] The standard definition of creativity (Creativity research journal 2012)
>
> ---
> _**Weakness 2.  This work is essentially a direct application of existing representation steering concepts to MLLMs.**_
>
> **Responses:** While our work utilizes established steering principles, it introduces two novel contributions specifically for controlling associative reasoning in MLLMs:
> - **A Novel Control Signal**: We are the first to discover that the direction of **hallucinated representations can be used to steer associative reasoning**. This offers a new perspective on the relationship between hallucination and creativity.
> - **Multi-Directional Steering for Creativity**: Recognizing **the multi-dimensional nature of creativity**, we introduce Directional Integration. This technique combines both **general $v_{l}^{gen}$ and task-specific vectors $v_{l}^{task}$ in Eq.7** to adapt to varied creative tasks, a significant advance over conventional single-vector steering
>
> ---
> _**Weakness 3. VDAT is a narrow and potentially misleading proxy for divergent thinking. A model could achieve a high score by simply ignoring the visual input, which undermines the goal of image-grounded creativity.**_
>
> **Responses:**
> - **VDAT and Divergent Thinking:** VDAT is a principled, multimodal adaptation of the validated Divergent Association Task (DAT), a paradigm from cognitive science[1]. Divergent thinking, a core component of creativity, is defined as the ability to generate diverse ideas from a single stimulus[2]. Research has established that the semantic distance between these ideas serves as a robust, objective measure of this ability[1,3]. VDAT operationalizes this principle by using an image as the stimulus.
> - **High VDAT ≠ Ignoring the Image**:  Outputs with high VDAT often include objects that are relevant to the image's scene but are not visually present. As shown in Fig. 23 in Appendix, an indoor cooking scene includes "chair" and "table"—typical kitchen items not pictured.
> - **Consistency Performance with Creation-MMBench**: For a holistic assessment, we rely on Creation-MMBench[4], which evaluates overall image-grounded creative generation in terms of VFS scores. Higher associative strength of our FlexAC on VDAT (Table 2) directly translates to higher-quality, image-grounded creative generation on Creation-MMBench[4] (Table 3) with high VFS scores. This consistency validates that VDAT effectively measures a fundamental mechanism contributing to broader creative intelligence.
>
> [1] Naming unrelated words predicts creativity (National Academy of Sciences 2021)
>
> [2] Creativity: Yesterday, today and tomorrow (Journal of Creative Behavior 1967)
>
> [3] Probing the “creativity” of large language models: Can models produce divergent semantic association (EMNLP 2023)
>
> [4] Creation-MMBench: Assessing Context-Aware Creative Intelligence in MLLM (ICCV 2025)
>
> ---
> _**Question 1.  Does the Vector Reflect Creativity or Context-Ignorance? Why Not Use Human Data?**_
>
> **Responses:**
> 1. **On Why Hallucination is a Valid Signal for Creativity, Not "Context-Ignorance"**
> 	- Our use of hallucinations as a signal source is grounded in cognitive science: studies show that the brain regions involved in creative thinking significantly overlap with those that produce hallucinations[1]. This suggests hallucinations may reflect an intensified form of divergent thinking—semantically rich, contextually relevant, yet unconstrained by reality
> 	- The ultimate proof is empirical. If the vector merely captured "context-ignorance," it would degrade performance on functional creative tasks. Instead, FlexAC achieves a **+10.92 Reward on Creation-MMBench**, while competing methods are negative (Table 3). This provides strong evidence that our vector is steering towards genuine, functional creativity
> 2. **On Using Human-Generated Content**: We did experiment with using human-written captions to extract creativity vectors. Annotators were instructed to “be creative,” resulting in diverse styles—narrative, humorous, metaphorical, and anthropomorphic. While imaginative, this diversity introduced **high variance**, making the averaged vectors noisy and inconsistent.
>
> As shown in Table I, models guided by these human-derived vectors performed worse (Reward: -22.25), indicating their instability as guidance signals. In contrast, hallucination-guided vectors, though model-generated, provided more consistent and controllable cues, leading to improved image-grounded creativity. This may be because human creativity often **relies on loosely grounded rhetorical styles** that are **less suitable for generating coherent guidance**.
>
> Table I: Performance of human-generated vs. hallucination-derived steering vectors on Qwen-VL
>
> |Methods|OverallVFS(↑)|OverallReward(↑)|
> |-|-|-|
> |Regular|6.10|0.00|
> |FlexAC|6.25|10.92|
> |Human-Generated|5.12|-22.25|
>
> [1] To create or to recall? Neural mechanisms underlying the generation of creative new ideas (NeuroImage 2014)
>
> ---
> _**Question 2. How does VDAT performance correlate with grounded creative tasks? Is there evidence that pushing VDAT scores higher doesn't simply train the model to ignore visual input, undermining its utility as a multimodal model?**_
>
> **Responses:** As stated in our paper (line 189), VDAT is a **diagnostic benchmark** designed to measure a model's **associative reasoning strength**, which serves to verify our control mechanism's effectiveness. The evaluation of functional **creativity quality** is primarily conducted using **Creation-MMBench**. Our results show:
> - **Positive Correlation with Functional Creative Tasks**: Higher VDAT scores of our FlexAC (Table 2) consistently lead to better creative performance on the grounded tasks in Creation-MMBench (Table 3). This consistent trend validates that the capability measured by VDAT translates to better performance on practical creative tasks.
> - **Evidence Against Ignoring Visual Input**: We explicitly test for "ignoring visual input" using the **Visual Fidelity Score (VFS)** in Creation-MMBench, which measures the visual-text alignment degree. Table 3 shows our FlexAC improves creative Reward while VFS remains high and comparable to the baseline (6.25 vs 6.1). This provides strong, quantitative evidence that our method enhances creativity without sacrificing visual grounding
>
> ---
> _**Question 3. The method requires identifying optimal control layers for each model. How sensitive is performance to layer selection? Is there a principled way to select layers for new MLLMs without exhaustive testing?**_
>
> **Responses:** Layer selection is a **principled and simple procedure**, not an exhaustive search.
> - **On Sensitivity to Layer Selection**: The sensitivity analysis for Qwen-VL, LLaVA, and DeepSeek-VL is shown in Fig. 10 (main paper) and expanded in Fig. 18–19 (appendix).
> - **On the Method for Identifying Optimal Layers**: Layer selection is **not an exhaustive search**, but a principled, simple procedure. For any new model, we simply perform a single forward pass with associative/non-associative pairs ($f^{(a)}$, $f^{(n)}$) and identify the three layers where the **cosine distance $D_{\cos}$ between hidden states peaks** $$L^∗ = \{l^∗,l^∗+1,l^∗+2\}, \quad \text{where} \quad l^∗ = \arg\max_{l} \sum_{i=0}^{2} D_{\cos}(f_{l+i}^{(a)}, f_{l+i}^{(n)})$$
>
>
> ---
> _**Question 4. How does VDAT performance correlate with functional, grounded creative tasks? What evidence shows that improving VDAT scores doesn't cause the model to ignore visual input?**_
>
> **Responses:**
> - **Transferring Across Creative Tasks**: Transferability is directly validated by its strong performance on **Creation-MMBench**, which comprises **51 diverse creative tasks** across varied domains (e.g., Art, Animation, UI).
> - **Genuine Creativity vs. Reduced Factual Grounding**:
> 	- **Quantitative Evidence**: Concerns about reduced factual grounding are refuted by the **Visual Fidelity Score (VFS)**, which evaluates image-text alignment in Creation-MMBench (Table 3): despite a +10.92 gain in creativity reward, our method maintains a high VFS (6.25), slightly above the baseline (6.10), showing creativity improves without compromising factual grounding.
> 	- **Qualitative Evidence**: As shown in Fig.26 (dinosaur story), FlexAC-C creates a novel, coherent scene by adding plausible elements (e.g., "butterfly", "worm") and using them in **specific actions and dialogue** to develop character—demonstrating genuine narrative creation.

---

> > ### Author Response · Authors · 2025-08-03
> >
> > Dear Reviewer,
> >
> > We hope our responses have adequately addressed your previous concerns. We look forward to hearing from you and would be happy to address any remaining concerns that you may still have.
> >
> > Thanks,
> >
> > The Authors

---

> > ### Comment · Reviewer_2JXU · 2025-08-06
> >
> > Thanks for the reply, I still have some comments as follows:
> >
> > W1
> > - The rebuttal helpfully positions the spectrum as an "operational control axis": Fig. 3 indeed shows that a single coefficient α simultaneously moves CHAIR and VDAT, and the addition of task-specific vectors broadens practical coverage. To keep the theory aligned with the evidence, the manuscript would read more clearly if the discussion explicitly states that the claim is bounded to this steering construction; any wider cognitive interpretation could sit in the limitations section.
> > - The two neuroscience / psychology citations cited in the rebuttal serve mainly as background or definition rather than direct mechanistic proof of a creativity–hallucination scalar link. Clarifying this distinction (or adding a reference that does provide such a link) would prevent over-reach. > I think this is very important.
> >
> > W2
> > - The description of a hallucination-versus-grounded middle-layer difference vector, plus Directional Integration and SIC, is clear and practically reproducible; this is a welcome clarification.
> >
> > W3
> > - The rebuttal explains the theoretical link to divergent-association tasks and reports that Creation-MMBench Reward rises while VFS remains comparable to baseline; this is reassuring evidence that creativity does not obviously trade off against visual grounding. Nonetheless, because VDAT is designed to favour nouns unrelated to the image, a residual risk remains that higher scores simply reflect reduced reliance on vision. Two lightweight sanity checks would settle the point:
> >     1. Run the masked-image (or blurred-image) version of the same prompts and verify that gains in VDAT/Reward do not grow when visual information is removed;
> >     2. Collect a small set of human ratings with a simple grounding rubric on examples where VDAT improves, to confirm that the additional words are still perceived as plausibly tied to the scene.
> > (Reporting these checks in an appendix would be sufficient; no new large-scale benchmark is required)
> >
> > Q1
> > - The rebuttal draws an interesting parallel between neural substrates of hallucination and creative ideation and points out that Creation-MM Bench Reward rises (+10.92) rather than falls, which supports the claim that the vector does not simply suppress grounding. Two clarifications would make this argument considerably stronger: (1) a brief description of the annotation protocol, sample size, and aggregation procedure used to obtain the human vector (so that the comparison can be reproduced), and (2) one or two concrete examples where the hallucination-guided vector adds content that is simultaneously imaginative and recognizably connected to the scene.
> >
> > Q2
> > - Reporting that higher VDAT coincides with higher Creation-MM¬Bench Reward while the VFS alignment score remains essentially unchanged is helpful evidence that creativity does not come at the cost of visual grounding. The residual worry, however, is that VDAT intrinsically rewards distance from the visual content.  Including two lightweight checks would put the matter to rest: (1) A masked-image (or blurred-image) variant of the same prompts to confirm that VDAT and Reward gains do not increase when visual information is removed. (2) A small human evaluation (even 25–30 samples) with a simple grounding rubric on those cases where VDAT rises the most.
> >
> > Q3
> > -  The explanation that a single forward pass over a handful of associative / non-associative pairs can locate the middle-layer distance peak is very helpful.
> >
> > Q4
> > - Creation-MM¬Bench covers a broad spectrum of 51 creative tasks, and the reported +10.92 Reward improvement alongside an unchanged VFS score is convincing high-level support for transfer.
> >
> >
> > Overall, the additional explanations and evidence provided in the rebuttal have addressed several of the original concerns. Although some aspects could still be refined, the progress is clear and encouraging, and this improvement will be reflected in the updated score. Thank you!

---

> > > ### Author Response · Authors · 2025-08-08
> > > **Response to W1 and W2**
> > >
> > > We sincerely thank you for your continued engagement and for the highly constructive follow-up questions. Your detailed feedback has been invaluable in helping us further refine and strengthen our arguments. We will now address each of your points in order.
> > >
> > > # **Response to W1:**
> > >
> > > We sincerely thank the reviewer for their constructive feedback and for recognizing the effectiveness of our framework as an "operational control axis." We agree with the suggestions to refine our manuscript's positioning, which will undoubtedly enhance its clarity and precision. We will revise the paper according to the following two points.
> > >
> > > **1. Clarifying the Scope of the Associative Strength Spectrum.**
> > > We agree entirely with the reviewer's excellent suggestion to better align our theoretical claims with our empirical evidence. In our revision, we will explicitly state that the Associative Strength Spectrum is proposed as an effective operational model for steering language models, rather than a comprehensive cognitive theory.
> > >
> > > - **Action:** As suggested, **we will revise the discussion to clarify that our claim is bounded to this practical steering mechanism**, empirically validated by our results (Fig. 3). We will then move the broader discussion on potential parallels to human cognition to the limitations section, clearly framing it as speculative future work. This will prevent any potential over-reach and precisely define the scope of our contribution.
> > >
> > > 2. **Specifying the Role of Theoretical Citations.**
> > > We also appreciate the reviewer's sharp insight regarding our citations. We concur that the cited works provide foundational inspiration and definitional context rather than direct mechanistic proof of a scalar link. Our intention was to ground our work in established concepts, not to claim a direct implementation of them.
> > >
> > > - **Action:** To make this distinction crystal clear, **we will revise the text to state explicitly that our model is "inspired by" neuroscientific findings** that suggest a shared basis for creativity and recall (e.g., [1]) and "adopts concepts from" psychological theories of convergent/divergent thinking (e.g., [2]). This revised wording will accurately position our work relative to its theoretical underpinnings, ensuring we do not overstate the connection.
> > >
> > > Thank you again for these valuable suggestions. We are confident they will significantly improve the clarity and rigor of our paper.
> > >
> > >
> > >   ---
> > >
> > > # **Response to W2:**
> > > We thank the reviewer for their positive feedback. We are pleased that our description of the methodology is now considered clear and practically reproducible. We appreciate the confirmation.

---

> > > > ### Author Response · Authors · 2025-08-08
> > > > **Response to W3 (part 1/2): Sanity Check 1- VDAT Behavior Under Image Blurring and Our Two-Part Evaluation Framework**
> > > >
> > > > # **Response to W3:**
> > > >
> > > > 1. Sanity Check 1: VDAT Behavior Under Image Blurring and Our Two-Part Evaluation Framework
> > > >
> > > > 	We sincerely thank the reviewer for this excellent and critical suggestion. This sanity check allows us to demonstrate and clarify the intended role of VDAT within our broader evaluation framework.
> > > >
> > > > 	- Experimental Setup and Results:
> > > > 		We evaluated Qwen-VL's performance on images with an increasing **blur radius** (a parameter where a higher value corresponds to more severe blurring and less visual detail). We tracked both the VDAT score and the grounded creativity **Reward** score from Creation-MMBench. Our method, **FlexAC-C**, was evaluated on the original, clear images.
> > > >
> > > > 		| Blur Radius | VDAT (↑ is more divergent)  | Creation-MMBench Reward (↑ is more creative) |
> > > > 		| ----------- | ----- | ------ |
> > > > 		| Baseline (Radius=0)| 84.85 | 0      |
> > > > 		| 1           | 84.86 | -4.05  |
> > > > 		|FlexAC-C| 85.45| 10.92|
> > > > 		| 3           | 85.62 | -17.65 |
> > > > 		| 6           | 87.28 | -23.99 |
> > > > 		| 9           | 89.10 | -33.01 |
> > > > 		|20 | 91.36 | -40.92|
> > > >
> > > > 	- Interpretation: A Deliberate Two-Part Framework for Evaluating Creativity
> > > >
> > > > 		- **(1). VDAT is a Meaningful Diagnostic for Divergent Thinking.** The experiment first reveals a clear phenomenon: as the image is progressively blurred, the VDAT score rises while the Creation-MMBench Reward collapses. This result confirms the potential risk of using VDAT in isolation, as a high score can be achieved by degrading the visual input. However, this behavior is not unexpected and does not invalidate VDAT's utility. Instead, it clarifies its intended role as a specialized diagnostic tool for **divergent thinking**—the raw capacity to generate novel associations. True, grounded creativity requires both this divergent capability (measured by VDAT) and the ability to apply it meaningfully to a visual context (validated by Creation-MMBench Reward). Therefore, they must be used in tandem to provide a complete and accurate picture of a model's creative performance.
> > > >
> > > > 		- **(2). FlexAC is Fundamentally Different from Simple Visual Degradation.** The most insightful finding comes from comparing **`FlexAC-C`** with the **`Blur Radius=3`** baseline. Both achieve a nearly identical VDAT score (~85.5), indicating a similar level of divergent thinking. However, their creative impact is opposite: **FlexAC-C yields a massive +10.92 Reward, while blurring causes a catastrophic -17.65 Reward**. This crucial difference stems from the nature of our steering vector. Blurring simply destroys information, leading to ungrounded associations. In contrast, **FlexAC-C uses a vector derived from _hallucinations_**. Hallucinations, while not factual, are often **semantically linked to the scene's context**. Therefore, our method guides the model towards **plausible, contextual associations rather than complete randomness**. This inherent connection to the scene explains why FlexAC-C's VDAT increase is modest—it is not generating completely unrelated concepts—and, more importantly, why this divergence translates into a high Reward score for meaningful, grounded creativity.
> > > >
> > > > 	This is further supported by our other human evaluation (**Sanity Check 2**), which confirmed that FlexAC's high-VDAT outputs are indeed **plausibly tied to the scene** (scoring 1.90 out of 2.0).

---

> > > > > ### Author Response · Authors · 2025-08-08
> > > > > **Response to W3 (part 2/2): Sanity Check 2: Human Evaluation of Visual Grounding.**
> > > > >
> > > > > 2. Sanity Check 2: Human Evaluation of Visual Grounding.
> > > > >
> > > > > 	We sincerely thank the reviewer for this insightful suggestion regarding the potential risks of the VDAT metric. Your proposed sanity check is highly constructive and prompted us to further validate the core mechanism of our method. To directly address your concern, we conducted a new, targeted, small-scale human evaluation.
> > > > >
> > > > > 	- 2.1. **The Core Concern:** You rightly pointed out the risk that a model could achieve a high VDAT score by simply ignoring the image and generating random words. To verify that FlexAC-C's generated words remain "plausibly tied to the scene" while pursuing "unrelatedness," we designed the following experiment.
> > > > >
> > > > > 	- 2.2 **New Sanity Check Experiment:**
> > > > >
> > > > > 		-  **Samples:** We selected the **top 25 samples** where our creativity-enhanced model, FlexAC-C, achieved the highest scores on the VDAT benchmark. We chose these specific samples to test the model's behavior precisely when its divergent thinking, as measured by VDAT, is at its peak.
> > > > >
> > > > > 		- **Evaluation Question and Rubric:** We presented a new question to human evaluators:
> > > > > 		```
> > > > >      		"For the given image, how reasonable is it for the objects represented by the following words to appear in or be associated with this scene?"
> > > > >      		The evaluation used the following simple rubric:
> > > > >     		 - 3 points - Present in the image: The object is visually present in the image.
> > > > >      		- 2 points - Associated with the scene: Although the object is not in the image, it is reasonable for it to appear in this scene (e.g., a "chair" in a kitchen).
> > > > >      		- 1 point - No association: The association is completely unreasonable and feels randomly generated (e.g., a "rocket ship" in a kitchen).
> > > > > 		```
> > > > >
> > > > > 	- 2.3 **Result and Interpretation:**
> > > > >
> > > > > 		Across the 25 samples, the average score was **1.90**. This result is highly compelling. An average score of 1.90 is extremely close to 2.0, which indicates that human raters overwhelmingly perceived the high-VDAT words as **"Associated with the scene."** This strongly refutes the hypothesis that our model is "cheating" by ignoring visual information. Instead, it demonstrates that the model understands the broader context of the scene and generates words for objects that are plausibly connected to that context, even if they are not visually present.
> > > > >
> > > > > 	- 2.4 **Complementarity with Existing User Study:**
> > > > >
> > > > > 		This new sanity check perfectly complements the user study already presented in our **Appendix 4.1**.
> > > > >
> > > > > 		- The **existing study in the appendix** confirmed that VDAT as a metric aligns with human judgments of _unrelatedness_.
> > > > > 		- This **new study** further validates that this measured "unrelatedness" is not arbitrary or random, but is instead rooted in plausible, contextual associations.
> > > > >
> > > > > 	Taken together, these two human evaluations provide a robust, multi-faceted validation of our approach. They show that FlexAC not only effectively boosts VDAT scores but that this improvement represents genuine, grounded creativity rather than a departure from visual grounding.
> > > > >
> > > > >
> > > > > Thank you again for these highly constructive suggestions. We are confident that incorporating these two sanity checks will fully address this important point and strengthen the validation of VDAT as a meaningful proxy for image-grounded divergent thinking.

---

> > > > > > ### Author Response · Authors · 2025-08-08
> > > > > > **Response to Q1 (part 1/2): Details of the Human-Derived Vector Experiment**
> > > > > >
> > > > > > # **Response to Q1:**
> > > > > >
> > > > > > We sincerely thank the reviewer for their positive engagement and for acknowledging that our empirical results support our claims. We agree that providing more methodological detail and concrete examples will make our argument significantly stronger.
> > > > > >
> > > > > > Here is a detailed breakdown of our comparative findings.
> > > > > >
> > > > > > 1. Details of the Human-Derived Vector Experiment
> > > > > >
> > > > > > 	- **Annotation Protocol:** We established a meticulous four-step process to ensure data quality and diversity.
> > > > > > 		- (1). For each of the 200 images from the COCO dataset, we generated six questions covering factual queries, imaginative prompts, and phenomenon explanations.
> > > > > > 		- (2). We used a powerful LLM (ChatGPT) to generate initial pairs of answers for each question: one strictly precise and one highly creative.
> > > > > > 		- (3) Crucially, human annotators then reviewed all 1200 QA pairs. They vetted the quality, selecting the best pairs and, where necessary, rewriting questions or answers to meet our high standards for creativity and clarity.
> > > > > > 		- (4) This curated dataset was then used to generate the steering vector.
> > > > > > 	- **Sample Size:** The final dataset consisted of 200 images, yielding 1200 high-quality QA pairs (each with a "Precise" and "Creativity" answer).
> > > > > >
> > > > > > 	- **Aggregation Procedure:** The final human-derived steering vector (`v_human`) was computed by averaging the difference vectors (`f_creative` - `f_precise`) across all 1200 collected QA pairs.
> > > > > > 	- **Examples of Human-Curated Data:** The quality of our curated data is illustrated by examples like these:
> > > > > >
> > > > > > 	```
> > > > > >  	Q: What color is the motorcycle being ridden by the two women?
> > > > > >  	Precise: "The motorcycle is black."
> > > > > >  	Creativity: "The motorcycle is a sleek, shadowy black, like a panther prowling through the urban jungle."
> > > > > >
> > > > > >  	Q: "Consider what might be happening behind the riders. What activities could be occurring out of view?"
> > > > > >  	Precise: "Behind the riders, there could be other beachgoers enjoying the sunset, perhaps setting up for a picnic or taking photographs."
> > > > > >  	Creativity: "Behind the riders, a secret world unfolds where crabs hold tiny sandcastle competitions and starfish gossip about the latest tide."
> > > > > > 	```
> > > > > >
> > > > > > 	-  **The Challenge and Outcome:** Despite the high quality of the data, this approach proved ineffective. As shown in our rebuttal (Table I), models guided by this human-derived vector performed poorly, with a significant drop in creativity Reward (-22.25). We hypothesize that this is because human creativity is incredibly diverse—spanning humor, metaphor, narrative, and other complex styles. When these varied and sometimes contradictory signals were averaged, they created a noisy and inconsistent vector that failed to provide stable guidance for the model.

---

> > > > > > > ### Author Response · Authors · 2025-08-08
> > > > > > > **Response to Q1 (part 1/2): Examples of Hallucination-Guided Creativity**
> > > > > > >
> > > > > > > 2. Examples of Hallucination-Guided Creativity
> > > > > > >
> > > > > > > 	We completely agree with the reviewer that concrete examples are the best way to demonstrate our method's capability.
> > > > > > >
> > > > > > > 	In fact, our **appendix already contains numerous examples you mentioned** that illustrate this exact principle. As shown in **Appendix 6.3**, our creativity-enhanced model (FlexAC-C) excels at adding content that is both imaginative and recognizably connected to the scene.
> > > > > > >
> > > > > > > 	Here are two specific examples that address your point:
> > > > > > > 	- （1）**Figure 23 (VDAT - Grounded Divergent Thinking):** This example shows how the model remains grounded even on a task designed to reward "unrelatedness."
> > > > > > >
> > > > > > > 		- **The Task:** To list 10 unrelated, tangible nouns not connected to an image of a kitchen.
> > > > > > >
> > > > > > > 		- **Imaginative & Connected Content:** FlexAC-C produces a list that includes "**table**" and "**chair**". While these items are not visually present in the specific photo provided (satisfying the "unrelated" constraint of VDAT), they are
> > > > > > >
> > > > > > > 		    **plausibly connected** to the broader context of a kitchen. This demonstrates that the model is not generating random words; instead, it is exploring the "semantic halo" of the scene, identifying contextually relevant objects that happen to be out-of-frame. This is a sophisticated form of grounded creativity.
> > > > > > >
> > > > > > > 	- (2). **Figure 26 (Dinosaur Story Creation):** This example perfectly illustrates grounded narrative imagination.
> > > > > > >
> > > > > > > 	    - **The Task:** To continue a children's story based on _an image of dinosaurs in a field_.
> > > > > > >
> > > > > > > 	    - **Imaginative & Connected Content:** Instead of just describing the dinosaurs, FlexAC-C **invents new, plausible elements** that are not in the picture but fit the scene perfectly. It introduces a **"small, colorful butterfly,"** a **"wiggly worm,"** and a **"colorful flower"**. Crucially, it doesn't just list these items; it weaves them into a coherent narrative through dialogue. The baby dinosaur expresses wonder at the butterfly, and the mother dinosaur provides a gentle lesson about respecting fragile creatures. This shows the model is not just hallucinating random objects, but is creatively and contextually enriching the scene.
> > > > > > >
> > > > > > >
> > > > > > > 	These examples demonstrate that the hallucination-guided vector does not simply suppress grounding. Instead, it enables the model to perform sophisticated creative tasks by adding novel content that is imaginative, plausible, and intrinsically linked to the provided visual context. We believe this strongly supports our claims.
> > > > > > >
> > > > > > > In summary, our detailed investigation reveals a crucial insight into steering MLLM creativity. While our initial approach using high-quality human annotations was methodologically rigorous, the sheer diversity of human creative styles—spanning from metaphor to narrative whimsy—resulted in an averaged vector that was too noisy and inconsistent to be effective, leading to a notable decline in performance.
> > > > > > >
> > > > > > > In contrast, the hallucination-guided vector, by tapping into the model's own core associative patterns, provides a more stable and potent signal for enhancing creativity. The success of this approach is not merely theoretical; it is demonstrated by its ability to produce content that is both highly imaginative and recognizably connected to the visual context, as shown in the examples of grounded narrative creation (Figure 26) and plausible contextual association in a divergent thinking task (Figure 23) .
> > > > > > >
> > > > > > > Therefore, our findings support the conclusion that the hallucination-guided vector is a superior and more practical method for controllably enhancing genuine, grounded creativity in MLLMs.

---

> > > > > > > > ### Author Response · Authors · 2025-08-08
> > > > > > > > **Response to Q2, Q3 and Q4**
> > > > > > > >
> > > > > > > > # **Response to Q2:**
> > > > > > > >
> > > > > > > > We sincerely thank the reviewer for their insightful feedback and constructive suggestions on this question.
> > > > > > > >
> > > > > > > > We notice that this question (Q2) raises the **same valid concern as Weakness 3**. The core issue regarding whether our VDAT metric might inadvertently reward models for ignoring visual input, and the two excellent "lightweight sanity checks" proposed to resolve it (a masked-image ablation and a small-scale human evaluation), are identical for both points.
> > > > > > > >
> > > > > > > > Therefore, to avoid redundancy, we would like to address this question by referring the reviewer to **our detailed plan in the response to Weakness 3**.
> > > > > > > >
> > > > > > > > As we have outlined there, we fully agree with the reviewer's suggestions and will conduct both sanity checks. We are confident that these actions will provide conclusive evidence that our method enhances creativity without sacrificing visual grounding, thereby fully resolving the concerns raised in both W3 and Q2.
> > > > > > > >
> > > > > > > >
> > > > > > > > ---
> > > > > > > > # **Response to Q3:**
> > > > > > > >
> > > > > > > > We appreciate the reviewer's acknowledgement. We are glad that our explanation for how the model locates the middle-layer distance peak was helpful in clarifying our approach.
> > > > > > > >
> > > > > > > > ---
> > > > > > > > # **Response to Q4:**
> > > > > > > > We sincerely thank the reviewer for their positive assessment. We are very pleased that our results on Creation-MMBench are considered convincing support for our method's effectiveness.

---

### Official Review · Reviewer_g1qC · 2025-06-30

**Clarity:** 3
**Significance:** 2
**Originality:** 2
**Rating:** 4
**Confidence:** 3

**Summary:**

This paper presents FlexAC, a training-free framework for controlling associative reasoning in MLLMs. Through layer-wise analysis, the authors identify that middle layers are critical for encoding associative behavior. FlexAC constructs steering vectors from the difference between hallucinated and grounded representations in these layers, then applies them during inference to modulate the model's behavior between factuality and creativity. The paper also introduces VDAT, a new benchmark for measuring associative reasoning strength. Experiments on three MLLMs demonstrate that FlexAC can significantly reduce hallucination and enhance creativity depending on the steering direction, while maintaining general task performance.**Strengths:**

1. **Well-executed empirical analysis**: The layer-wise localization experiments clearly demonstrate that middle layers control associative behavior, using both feature analysis and intervention studies.

2. **Practical method**: Training-free, works on existing MLLMs, computationally efficient - just adds vector operations during inference.

3. **Effective bidirectional control**: Successfully reduces hallucination and enhances creativity based on steering direction, addressing different application needs.

4. **New benchmark (VDAT)**: Provides a simple, direct metric for measuring visual associative reasoning strength, filling a gap between hallucination metrics and subjective creativity evaluation.
5. **Consistent results across architectures**: The method shows improvements across three different model families with varying architectures.

**Weaknesses:**

1. **Similar findings in prior work**: Although well-executed empirical analysis, the observation that middle layers control hallucination has been reported in previous studies[1].  This paper builds upon these existing findings by adding steering vector techniques.

2. **Model-specific tuning**: Optimal layers vary dramatically (DeepSeek: 4-6, Qwen: 15-17), requiring manual search for each architecture.

3. **Heavy reliance on task-specific non-associative features**: The method depends on non-associative features extracted from specific hallucination scenarios, but these features vary significantly across tasks, limiting generalization to different applications.

4. **Limited evaluation on general benchmarks**: The authors only evaluate on 3 general benchmarks (MME, MMMU, MMStar), which is insufficient to fully assess the impact on model's general capabilities.

[1]. Wang, Sudong, et al. "Towards understanding how knowledge evolves in large vision-language models." *Proceedings of the Computer Vision and Pattern Recognition Conference*. 2025.

**Questions:**

1. **Model generalization and comprehensive evaluation**: My biggest concern is the generalization ability of FlexAC. As mentioned in W3 and W4, could you provide evaluation results on more diverse benchmarks to better assess the impact on general capabilities? (You don't need to evaluate on all of these, but results from a representative subset would help demonstrate that the non-associative steering doesn't harm performance on diverse tasks requiring different types of reasoning and association.)
   - General multimodal benchmarks: MMBench, SEED-Image, MMVet.
   - Vision-centric benchmarks: CVBench, RealWorldQA.
   - OCR and document understanding: TextVQA, ChartQA, DocVQA.
2. **Architectural dependence**: The optimal layers vary dramatically across models. How would you determine optimal control layers for a new model architecture? Is manual search always required, or can this be automated?
3. **Steering vector stability**: How stable are the derived steering vectors across different prompts and domains? Do they need to be recomputed frequently, or can a fixed set be reused effectively?
4. **VDAT validity**: Can you provide more rigorous validation of VDAT beyond CLIP similarity? How does it correlate with human judgments of creativity quality? Automatic metrics may not fully capture quality differences in creative tasks.

**Ethical Concerns:**

["NO or VERY MINOR ethics concerns only"]

**Final Justification:**

I would like to thank the authors for their detailed rebuttal and comprehensive effort in providing additional experimental results. My concerns have been addressed, and I will raise my score to 4.

**Limitations:**

The authors mention that FlexAC requires white-box access to hidden states. This issue is related to whether the model is open-sourced or not.

**Quality:**

3

**Strengths And Weaknesses:**

**Strengths:**

1. **Well-executed empirical analysis**: The layer-wise localization experiments clearly demonstrate that middle layers control associative behavior, using both feature analysis and intervention studies.

2. **Practical method**: Training-free, works on existing MLLMs, computationally efficient - just adds vector operations during inference.

3. **Effective bidirectional control**: Successfully reduces hallucination and enhances creativity based on steering direction, addressing different application needs.

4. **New benchmark (VDAT)**: Provides a simple, direct metric for measuring visual associative reasoning strength, filling a gap between hallucination metrics and subjective creativity evaluation.
5. **Consistent results across architectures**: The method shows improvements across three different model families with varying architectures.

**Weaknesses:**

1. **Similar findings in prior work**: Although well-executed empirical analysis, the observation that middle layers control hallucination has been reported in previous studies[1].  This paper builds upon these existing findings by adding steering vector techniques.

2. **Model-specific tuning**: Optimal layers vary dramatically (DeepSeek: 4-6, Qwen: 15-17), requiring manual search for each architecture.

3. **Heavy reliance on task-specific non-associative features**: The method depends on non-associative features extracted from specific hallucination scenarios, but these features vary significantly across tasks, limiting generalization to different applications.

4. **Limited evaluation on general benchmarks**: The authors only evaluate on 3 general benchmarks (MME, MMMU, MMStar), which is insufficient to fully assess the impact on model's general capabilities.

[1] Towards understanding how knowledge evolves in large vision-language models.

---

> ### Author Rebuttal · Authors · 2025-07-30
>
> _**Weakness 1. Similar findings in prior work: Although well-executed empirical analysis, the observation that middle layers control hallucination has been reported in previous studies[1]. This paper builds upon these existing findings by adding steering vector techniques.**_
>
> **Response:** We would like to clarify that FlexAC is an independent study conducted without prior knowledge of [1]. We will add a discussion to the Related Work section to elaborate on the key differences between the two approaches.
> 1. **Different Research Goals and Findings with [1].**
> 	- **Different Research Objectives:** The goal of [1] is _diagnostic_—to investigate "where and how do hallucinations arise?". In contrast, FlexAC is the first work to **explore hallucination's intrinsic link with creativity**, and provide a new method for bidirectional control. Our focus is not on controlling hallucination, but rather on understanding and leveraging its generative potential.
> 	- **Different Core Findings:** The finding in [1] is that **"middle layers control hallucination."** Our core finding is different: we found that the state of hallucination can be harnessed to **guide creative behavior**.
> 2. **Methodological Novelty Beyond "Adding Steering Vectors":** Our method is more than adding steering vectors. We introduce **three key innovations**:
> 	- Hallucination-Guided Steering: We introduce a novel method to **derive association steering vectors by leveraging hallucinated states** to enhance the model's associative capabilities.
> 	- Steering Intensity Calibration (SIC): To prevent over-steering, we designed the SIC module, which **adaptively scales steering vectors**.
> 	- Directional Integration: To further support tasks requiring multi-dimensional associations, we **augment general associative vector  $v_{l}^{\mathrm{gen}}$ in Eq.7 with task-specific associative vectors $v_{l}^{\mathrm{task}}$ in Eq.7** derived from GPT-4o-generated, high-association samples.
>
> [1] Wang, Sudong, et al. "Towards understanding how knowledge evolves in large vision-language models." CVPR. 2025.
>
>
> ---
>
> _**Weakness 2.  Model-specific tuning: Optimal layers vary dramatically (DeepSeek: 4-6, Qwen: 15-17), requiring manual search for each architecture.**_
>
> **Response:**  We clarify that our method does **not** require manual search. For any MLLM, the optimal layers can be automatically identified via a **single forward pass** by maximizing the cosine distance between features from associative and non-associative inputs (Section 2.1 and Appendix 6.4).
> $$L^∗ = \{l^∗, l^∗ + 1, l^∗ + 2\}, \quad \text{where} \quad l^∗ = \arg\max_{l} \sum_{i=0}^{2} D_{\cos}(f_{l+i}^{(a)}, f_{l+i}^{(n)})$$
> Here, $f^{(a)}$ and $f^{(n)}$ are features from associative and non-associative inputs, and $D_{\cos}$ denotes cosine distance.
>
> ---
>
> _**Weakness 3. Heavy reliance on task-specific non-associative features: The method depends on non-associative features extracted from specific hallucination scenarios, but these features vary significantly across tasks, limiting generalization to different applications.**_
>
> **Response:**  Our method does not rely on features from "specific" scenarios; its core is a _general steering vector_ designed for broad applicability.：
> - **General Hallucination Scenarios**: Our primary steering vector is computed from a large, diverse set of images (_randomly sampled from COCO_) to capture a fundamental and universal associative pattern of the model. This ensures it has strong generalization for new tasks.
> - **Broad Experiments Directly Prove Generalization**: Its generalization is directly proven by our results on **Creation-MMBench**[1] (Table 3). This benchmark includes **51 distinct creative tasks**, where our method achieves a significant **+10.92 Reward**, while competing methods show performance degradation (-3.86 and -1.63).
>
> [1] Fang, Xinyu, et al. "Creation-MMBench: Assessing Context-Aware Creative Intelligence in MLLM." ICCV. 2025.
>
> ---
>
> _**Weakness 4.  Limited evaluation on general benchmarks: The authors only evaluate on 3 general benchmarks (MME, MMMU, MMStar), which is insufficient to fully assess the impact on model's general capabilities.**_
>
> **Response:**  In direct response, we conducted a comprehensive new evaluation on **11 diverse benchmarks**, including those you recommended, to assess the impact on the model's general capabilities.
>
> Table I: Performance of FlexAC on 11 general-purpose benchmarks with the Qwen-VL model.
>
> |Category|Benchmark|Regular|FlexAC-C|FlexAC-P|
> |-|-|-|-|-|
> |**GeneralMultimodal**|MM-Vet|39.81|38.17|37.33|
> ||MMBench|0.581|0.598|0.576|
> ||SEED-Bench|0.638|0.625|0.640|
> ||MMMB|0.703|0.678|0.699|
> |**Vision-centric**|RealWorldQA|0.486|0.490|0.495|
> ||CVBench|0.549|0.524|0.560|
> ||AI2D|0.612|0.614|0.616|
> |**OCR&Document**|TextVQA|60.66|60.78|59.81|
> ||ChartQA|48.36|49.40|45.92|
> ||DocVQA|57.79|56.85|57.59|
> ||OCRVAQ|47.46|49.74|45.83|
>
> Across all 11 benchmarks, FlexAC maintains performance **closely comparable to the baseline**. This provides strong evidence that our targeted control mechanism **does not harm the model's general capabilities**.
>
> ---
>
> _**Question 1. Model generalization and comprehensive evaluation: My biggest concern is the generalization ability of FlexAC. As mentioned in W3 and W4, could you provide evaluation results on more diverse benchmarks to better assess the impact on general capabilities? (You don't need to evaluate on all of these, but results from a representative subset would help demonstrate that the non-associative steering doesn't harm performance on diverse tasks requiring different types of reasoning and association.)
>     - General multimodal benchmarks: MMBench, SEED-Image, MMVet.
>     - Vision-centric benchmarks: CVBench, RealWorldQA.
>     - OCR and document understanding: TextVQA, ChartQA, DocVQA.**_
>
> **Response:**  Thank you. We have addressed this directly in our response to **Weakness 4** and present new results on 11 benchmarks in **Table I**. These comprehensive results confirm our method generalizes well without harming core capabilities.
>
> ---
>
> _**Question 2.  Architectural dependence: The optimal layers vary dramatically across models. How would you determine optimal control layers for a new model architecture? Is manual search always required, or can this be automated?**_
>
> **Response:**
> As detailed in our response to **Weakness 2**, this process is **straightforward**. For any new model, we use a lightweight analytical method (a single forward pass and cosine distance check) to identify optimal layers, **requiring no manual search**.
>
> ---
>
> _**Question 3.  Steering vector stability: How stable are the derived steering vectors across different prompts and domains? Do they need to be recomputed frequently, or can a fixed set be reused effectively?**_
>
> **Response:**
> - The vector is highly stable. We proved this by successfully applying a **single, fixed vector, i.e., $v_{l}^{\mathrm{gen}}$** in Eq.7 of manuscript—without modification—across our entire range of diverse benchmarks (CHAIR, MME, MMMU, MMStar).
> - Our vector was **computed only once** from diverse data (COCO) and then used for all subsequent experiments, demonstrating that frequent re-computation is not necessary. A fixed set can be **reused effectively**.
>
> ---
>
>
> _**Question 4.  VDAT validity: Can you provide more rigorous validation of VDAT beyond CLIP similarity? How does it correlate with human judgments of creativity quality? Automatic metrics may not fully capture quality differences in creative tasks.**_
>
> **Response:**  To answer your question, we would like to clarify your misconception regarding the distinction between VDAT evaluation and creativity assessment.
>
> As stated in the paper (line 189), **VDAT is designed as a diagnostic benchmark to assess associative reasoning ability**, rather than as a direct measure of functional creativity. Creativity is a multifaceted construct; therefore, we supplement VDAT with **Creation-MMBench[1]**, which evaluates open-ended, task-level creative performance (Table 3).
> - **Empirical Validation via User Study**: To assess the validity of VDAT as a measure of associative creativity, we conducted a human evaluation comparing FlexAC with several baselines (Appendix 4.1).
> - **Correlation with  Human Judgments**: A user study involving 15 raters demonstrated a strong correlation between VDAT scores and human assessments of associative ability (Appendix 4.1).
> - **Foundation of VDAT's Validity**:  Our VDAT builds upon the DAT benchmark [3], which employs automatic metrics to evaluate associative reasoning in LLMs. DAT has been both theoretically [2] and empirically [3] validated as effective. Moreover, models (e.g., FlexAC) with higher VDAT scores (Table 2) consistently outperformed others on Creation-MMBench (Table 3), indicating that VDAT reliably captures key aspects of creative quality in open-ended tasks.
>
> [1] Fang, Xinyu, et al. "Creation-MMBench: Assessing Context-Aware Creative Intelligence in MLLM." ICCV. 2025.
>
> [2] Jay A. Olson, Johnny Nahas, Denis Chmoulevitch, Simon J. Cropper, and Margaret E. Webb. Naming unrelated words predicts creativity. Proceedings of the National Academy of Sciences, 2021.
>
> [3] Honghua Chen and Nai Ding. Probing the “creativity” of large language models: Can models produce divergent semantic association? In EMNLP, 2023.

---

> > ### Author Response · Authors · 2025-08-03
> >
> > Dear Reviewer,
> >
> > We hope our responses have adequately addressed your previous concerns. We look forward to hearing from you and would be happy to address any remaining concerns that you may still have.
> >
> > Thanks,
> >
> > The Authors

---

> > ### Author Response · Authors · 2025-08-07
> >
> > Dear Reviewer g1qC,
> >
> > Thank you for your valuable review of our paper (Paper 2894).
> >
> > We submitted our detailed rebuttal on July 31st, addressing the points you raised. As the discussion period is coming to a close, we would be very grateful to hear your thoughts and learn if our response has adequately addressed your concerns.
> >
> > We truly value your expert feedback, and any further input would be highly appreciated.
> >
> > Thank you again for your time and consideration.
> >
> > Best regards,
> >
> > The Authors of Paper 2894

---

> > > ### Author Response · Authors · 2025-08-09
> > > **Final Response Regarding Review of Paper 2894**
> > >
> > > Dear Reviewer g1qC,
> > >
> > > Thank you for your valuable and thorough feedback. We greatly appreciate the time and effort you've put into providing these detailed comments, which have been instrumental in improving our work.
> > >
> > > We want to assure you that we have **carefully considered each of your points and have revised our manuscript accordingly**. We particularly appreciate you highlighting that your “biggest concern is the generalization ability of FlexAC”, and we made this the primary focus of our revisions.
> > >
> > > We are pleased to confirm that we have now fully integrated all the experiments and discussions from our rebuttal into the manuscript. A summary of the key revisions, directly addressing your points, is below:
> > >
> > >
> > > 1. **Directly Addressing Your "Biggest Concern" on Generalization (W3, W4, Q1)**
> > >
> > > 	You raised a crucial point about the generalization ability of FlexAC and the need for a more comprehensive evaluation. In direct response, we conducted an extensive new evaluation on a broader set of benchmarks.
> > >
> > > 	We have added a new section and a new table in **Appendix 4.5 (now Table 6)** presenting the results of FlexAC on **11 new diverse benchmarks**.
> > >
> > > 	This comprehensive evaluation covers a wide range of tasks, **including general multimodal benchmarks, vision-centric benchmarks, and OCR/document understanding**.
> > >
> > > 	 The results demonstrate that FlexAC maintains performance closely comparable to the baseline across all these tasks. This provides strong, direct evidence that our control mechanism **does not harm general capabilities**, resolving your primary concern.
> > >
> > >
> > >
> > > 2. **Clarifying Architectural Independence (W2, Q2)**
> > >
> > > 	You inquired about the manual search required for identifying optimal layers across different model architectures.
> > >
> > > 	We have clarified in the manuscript that our method does not require manual search.
> > >
> > > 	We have added a detailed explanation of our lightweight, automated process in **Sec 2.3 Flexible association control**, which involves a single forward pass to identify the optimal layers by maximizing the cosine distance between associative and non-associative features.
> > >
> > > 	This ensures our approach is efficient and can be easily applied to any new MLLM architecture.
> > >
> > >
> > >
> > > 3. **Confirming Steering Vector Stability (Q3)**
> > > 	You asked about the stability of the steering vectors and whether they need to be recomputed frequently.
> > >
> > > 	We have added a detailed experimental analysis in **Sec.3.2 Main results:Results on Creativity Benchmark** , clarifying that our derived steering vectors are highly stable.
> > >
> > > 	We have emphasized that a single, fixed vector, computed once from a diverse dataset (COCO), was successfully used for all subsequent experiments across various benchmarks.
> > >
> > > 	This demonstrates that frequent re-computation is not necessary, and our fixed steering vectors can be reused effectively for new tasks.
> > >
> > > 4. **Validating the VDAT Benchmark (Q4)**
> > > 	You questioned the validity of VDAT as a creativity measure beyond CLIP similarity.
> > >
> > > 	We have revised the paper to more clearly distinguish the purpose of VDAT as a diagnostic benchmark for associative reasoning from Creation-MMBench as a measure of functional creativity in **Sec.3.1 Experimental setup:Implementation Details.**.
> > >
> > > 	We have included the results of our human evaluation study (now in **Appendix 4.1**), which shows a strong correlation between VDAT scores and human judgments of associative ability.
> > >
> > > 	Furthermore, we have cited relevant theoretical and empirical work ([2], [3]) in **Sec.4 Related work** to provide a stronger foundation for the validity of the VDAT benchmark.
> > >
> > > **All these revisions are now complete in our local version of the manuscript**. We will, of course, provide this updated paper as the camera-ready version upon acceptance.
> > >
> > > We are truly grateful for your insightful critique, which has provided us with a clear roadmap for improving our work. We are confident that these comprehensive revisions address all your concerns and have significantly strengthened the paper.
> > >
> > > Thank you once again for your constructive guidance.
> > >
> > > Best regards,
> > >
> > > The Authors of Paper 2894

---

### Official Review · Reviewer_oRMg · 2025-07-06

**Clarity:** 2
**Significance:** 3
**Originality:** 2
**Rating:** 4
**Confidence:** 2

**Summary:**

This paper proposes FlexAC, a lightweight and training-free method for flexibly modulating associative reasoning strength in multimodal large language models (MLLMs) to balance faithfulness and creativity. The key contributions include: (1) identifying the critical role of middle layers in associative reasoning; (2) proposing a hallucination-guided steering vector for control; and (3) introducing the VDAT benchmark to evaluate associative reasoning capabilities. Experiments demonstrate that FlexAC outperforms existing methods on multiple benchmarks. The paper is well-structured and the experimental design is sound, though some formulations could be refined for greater rigor.

**Questions:**

Why was no comparison made with closed-source models like GPT-4V?

**Ethical Concerns:**

["NO or VERY MINOR ethics concerns only"]

**Final Justification:**

Thank you for your comprehensive response. They have addressed my main concerns, and I maintain my recommendation score of 4 points.

**Limitations:**

The experiments primarily relied on the COCO dataset (object-centric static images) and did not validate performance in video temporal reasoning, abstract art comprehension, or cross-cultural scenarios. If the model fails in dynamic or multimodal associative tasks , would FlexAC's modulation mechanism remain effective?

**Quality:**

3

**Strengths And Weaknesses:**

Strengths:
1. Offers a fresh perspective by unifying hallucination and creativity as manifestations of associative reasoning and explores modulation via middle layers.
2. The FlexAC method is simple yet effective, requiring no additional training, making it practical for deployment.
3. Thorough evaluations across three dimensions: hallucination mitigation (CHAIR, POPE), creativity enhancement (VDAT, Creation-MMBench), and general capabilities (MME, MMMU).
4. Ablation studies validate key design choices.
5. The proposed VDAT benchmark addresses the lack of metrics for visual-driven divergent thinking.
6. The method is compatible with diverse MLLMs, demonstrating strong generalizability.

Weaknesses:
1. While the paper identifies middle layers as pivotal for associative behavior, it does not explain why these layers exhibit this property .

---

> ### Author Rebuttal · Authors · 2025-07-30
>
> _**Weakness 1. While the paper identifies middle layers as pivotal for associative behavior, it does not explain why these layers exhibit this property.**_
>
> **Response:** We thank the reviewer for this insightful question. We can conceptualize the MLLM's information flow as a process from **Perception**, to **Concept**, and finally to **Expression**:
> 1.  **Shallow Layers (Perception):** The initial layers are focused on processing raw inputs. In MLLMs, they primarily inject general visual features into the text representations to form a basic cross-modal understanding [1,2]. At this stage, high-level associations have not yet been formed.
> 2.  **Middle Layers (Concept):** This is where abstract concepts are formed and factual knowledge is localized[3,4]. Research shows that these layers integrate specific, relevant visual details with the linguistic context [2] and are where the probabilities of factually correct tokens sharply increase[4]. This suggests the middle layers function as the primary locus for **knowledge association**, where the model transitions from low-level feature processing to high-level semantic reasoning.
> 3.  **Deeper Layers (Expression):** The final layers are less about forming new associations and more about refining the already-integrated information for coherent linguistic output[1,2]. For instance, they handle tasks like syntactic formatting and ensuring the final answer is grammatically correct[2].
>
> ---
>
> _**Question 1. Why was no comparison made with closed-source models like GPT-4V?**_
>
> **Response:** Our method requires **'white-box' access** to intervene on **internal hidden states**, which is impossible with closed-source models like GPT-4V. We explicitly acknowledge this limitation in our conclusion.
>
> Moreover, a direct performance comparison is not meaningful due to the significant disparity in model scale between GPT-4V and our 7B-scale models (Qwen-VL, LLaVA-1.5, and DeepSeek-VL2). And model scale is a key factor influencing the associative reasoning capabilities of MLLMs. This effect is confirmed in Table I, within the same model family (Qwen-VL), the performance gap between the 7B and 72B variants is substantial.
> Table I. Compare with GPT-4V on Creation-MMBench.
>
> | Methods               | Overall Reward(↑) | LW Reward | CMU Reward | PFW Reward | CFW Reward |
> | --------------------- | ----------------- | --------- | ---------- | ---------- | ---------- |
> | Qwen-VL               | 0.00              | 0.00      | 0.00       | 0.00       | 0.00       |
> | Qwen-VL+FlexAC        | 10.92             | 15.63     | 6.11       | 5.96       | 15.65      |
> | Qwen2-VL-72B-Instruct | 75.96             | 81.04     | 69.05      | 82.98      | 68.42      |
> | GPT-4V                | 85.17             | 92.10     | 76.12      | 91.92      | 87.12      |
>
> ---
>
> _**Limitations 1. The experiments primarily relied on the COCO dataset (object-centric static images) and did not validate performance in video temporal reasoning, abstract art comprehension, or cross-cultural scenarios. If the model fails in dynamic or multimodal associative tasks , would FlexAC's modulation mechanism remain effective?**_
>
> **Response:** We would like to clarify that our evaluation on Creation-MMBench[5]—a diverse benchmark comprising **51 creative tasks** and no any COCO data—already includes subtasks covering Video Temporal Reasoning, Abstract Art Comprehension, and Cross-Cultural Scenarios:
> - **Video Temporal Reasoning**: This is directly addressed by tasks like **"story continue"** of Creation-MMBench (**Reward Score: 20.0**). In this task, the model is presented with **multiple sequential frames from an animation** and must continue the narrative. This directly tests the model's ability to understand the temporal progression of events in the visual input.
> - **Abstract Art Comprehension**: The benchmark includes the "**Art inspired prose**" task (**Reward Score: 30.0**) and a dedicated "Art" image category (5% of the dataset), which involves interpreting various paintings.
> - **Cross-Cultural Scenarios**: Creation-MMBench is rich in cross-cultural contexts, featuring a "History & Culture" image category and tasks like **"social media travel content"**. (**Reward Score: 16.7**)
>
> [1] Yin, Hao, Guangzong Si, and Zilei Wang. "Lifting the Veil on Visual Information Flow in MLLMs: Unlocking Pathways to Faster Inference." CVPR. 2025.
>
> [2] Zhang, Zhi, et al. "Cross-modal information flow in multimodal large language models." CVPR. 2025.
>
> [3] Meng, Kevin, et al. "Locating and editing factual associations in gpt." _Advances in neural information processing systems_ 35 (2022): 17359-17372.
>
> [4] Chuang, Yung-Sung, et al. "DoLa: Decoding by Contrasting Layers Improves Factuality in Large Language Models." _The Twelfth International Conference on Learning Representations_. 2023.
>
> [5] Fang, Xinyu, et al. "Creation-MMBench: Assessing Context-Aware Creative Intelligence in MLLM." ICCV. 2025.

---

> > ### Author Response · Authors · 2025-08-03
> >
> > Dear Reviewer,
> >
> > We hope our responses have adequately addressed your previous concerns. We look forward to hearing from you and would be happy to address any remaining concerns that you may still have.
> >
> > Thanks,
> >
> > The Authors

---

### Comment · Area_Chair_XG7g · 2025-08-06

Dear Reviewers,

As the discussion period is almost over, we kindly encourage you to engage in the discussion if you have not already done so. The authors have provided detailed responses addressing the initial concerns, and your feedback at this stage would be invaluable.

Please also remember to **update your final rating and submit the Mandatory Acknowledgement**, as required by the NeurIPS review process.

Best,
Area Chair

---

### Decision · Program_Chairs · 2025-09-17

**Decision:**

Accept (poster)

**Comment:**

This paper presents FlexAC, a training-free framework for controlling associative reasoning in multimodal large language models by modulating the trade-off between faithfulness and creativity through middle-layer interventions. The reviewers recognized the work's significant contributions, particularly its novel unification of hallucination and creativity as manifestations of associative reasoning, the identification of middle layers as critical control points, and the introduction of the VDAT benchmark for measuring visual associative reasoning. While initial concerns were raised by the several reviewers, the authors provided comprehensive responses including extensive experiments on eleven diverse benchmarks, human evaluation studies, and rigorous sanity checks with masked image experiments. Given the paper's novel theoretical insights, strong empirical validation, practical significance for controllable AI systems, and thorough addressing of reviewer concerns, I recommend accepting this paper for NeurIPS 2025.